# DELTA-MIA: MEASURING MEMBERSHIP INFERENCE ATTACKS IN LARGE LANGUAGE MODELS VIA SELF-CONTRAST FRAMEWORK

## ABSTRACT

Membership inference attack (MIA) underpins privacy risk assessment, provenance, and compliance for large language models (LLMs). Observational evaluations confound membership with distribution shift, hide sample-level behavior, and assume access to proprietary corpora. We present Delta-MIA, an interventional self contrast framework that isolates genuine membership signals by comparing a model before and after controlled exposure to the same dataset. The pipeline records pre exposure responses on verifiably unseen data, performs full-parameter fine tuning on that data followed by stabilization, and computes sample level deltas. We introduce three diagnostics: explained variance ratio (EVR), mean vertical distance (MVD), and above diagonal ratio (ADR), which quantify noise, separation, and baseline detectability. Re-evaluating 9 MIA methods, several remain robust once shift is removed, while others such as DC-PDD and ConReCaLL decline markedly; Min K%++ shows strong separation with high MVD. Delta-MIA enables bias-free, interpretable, and transferable evaluation for MIA in LLMs.

## 1 INTRODUCTION

In the era of large language models (LLMs), membership inference attack (MIA) determines whether a given sample took part in training of a target model and serves as a core technique for privacy risk assessment, data provenance, and compliance auditing (Shokri et al., 2017; Shi et al., 2024). Accurate quantification of memorization risk enables effective governance strategies (Mireshghallah et al., 2022; Fry, 2025). Meanwhile, emerging needs such as machine unlearning (Geng et al., 2025) motivate industry and regulators to seek external verification methods that do not rely on the internal state of a model. These factors create a shared demand for *a reliable, interpretable, and transferable evaluation paradigm for MIA* in both research and practice.

However, as shown in Fig. 1 (a), existing evaluation pipelines for MIA in the LLM setting follow an observational paradigm. These pipelines exhibit three fundamental limitations that reduce the validity and credibility of evaluation results. **(1) Data distribution shift**: Recent studies construct the non-member set with data from time periods (Shi et al., 2024; Meeus et al., 2024) or domains different from the training data (Chalkidis et al., 2023). This practice creates differences not only in exposure to the model but also in underlying distributions. Such construction introduces confounding variables and enables attackers or models to exploit distributional shortcuts rather than genuine membership signals, which inflates performance under blind evaluation and distorts the measured capability of MIA methods (Meeus et al., 2025b). **(2) Coarse-grained analysis**: Mainstream evaluations rely on dataset-level aggregate metrics such as AUC computed on a static pretrained model (Carlini et al., 2021; Song & Mittal, 2020). This design obscures behavior at the sample level and prevents rigorous before–after comparison on the same model. As a result, key hypotheses about membership signals remain difficult to test in a falsifiable manner. **(3) Poor transferability**: Current frameworks implicitly require access to the training corpus of the target model or a specific partition strategy (Shi et al., 2024). For frontier LLMs, training data are proprietary and unavailable (Hardinges et al., 2024). This constraint hinders transferability across models and slows progress toward a unified and reproducible benchmark.

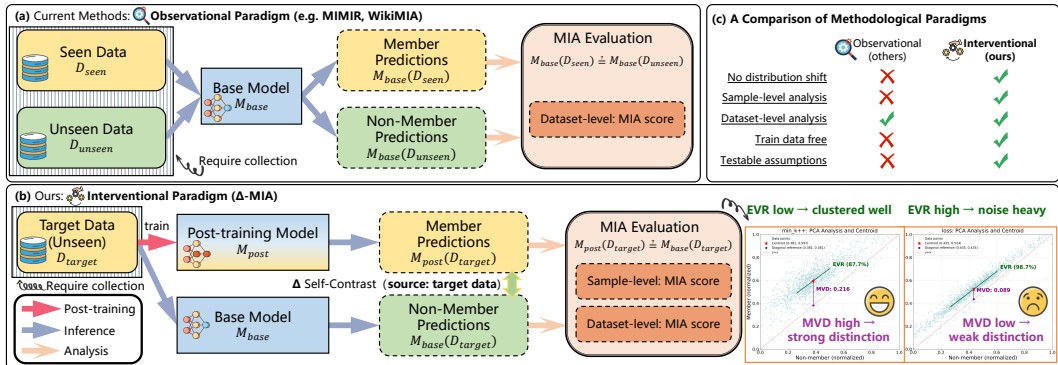

Figure 1: Schematic comparison of observational and interventional paradigms for MIA benchmarking. (a) **Observational Paradigm**: Relies on independent member/non-member datasets, risking distribution shift. Analysis restricted to dataset-level scores and requires original training data. (b) **Interventional Paradigm ($\Delta$-MIA)**: Fine-tunes a base model on target data. The same data serves as members for the post-training model and non-members for the base model, enabling sample-level analysis. Key indicators (inset): EVR (explained variance ratio of the first principal component) measures scatter clustering, while MVD (mean vertical distance to the diagonal) quantifies membership distinguishability. (c) **Paradigm Comparison**: The interventional design eliminates training data dependency, resolves distribution shift, and supports more granular, reliable MIA evaluation.

Several benchmarks within the observational paradigm attempt to mitigate data distribution shift. For example, MIMIR (Duan et al., 2024) constructs member and non-member sets by randomly splitting open-source LLM datasets to address temporal shift, and utilize n-gram overlap for deduplication and as a proxy to gauge potential shift. Although this approach reduces the issue, n-gram overlap captures only lexical-level differences rather than semantic ones, does not account for results on GitHub datasets where n-gram overlap is high while MIA performance remains strong, and may miss subtler distributional differences (Meeus et al., 2025b).

To address these challenges in a systematic way, we introduce Delta-MIA ($\Delta$-MIA), an MIA evaluation framework under an Interventional Paradigm, as shown in Fig. 1 (b). The core idea is to isolate and measure genuine membership signals by self-comparison of the model before and after exposure to the same data distribution. This procedure follows a **self-contrast** principle in which each sample acts as a non-member probe in the pre-exposure state and as a member probe after exposure, so that observed differences reflect membership effects. The framework follows three steps. (1) Collect a set of unseen data with verifiable non-overlap with the training corpus of the target model, and record responses in the pre-exposure state. (2) Apply incremental training using this data and subsequent training corpus for stabilization, to obtain the post-exposure state. (3) On the identical dataset, compute the delta between pre- and post-exposure responses to construct discriminative signals at the sample level. This interventional design keeps the distribution fixed and varies only exposure of the model, which removes confounders introduced by cross-domain or temporal sampling.

This interventional, self-contrast design yields three advantage. **(1) Bias-free evaluation**: It defines non-member and member roles via the change of model states before and after exposure, rather than by constructing a separate non-member set. This removes data distribution shift at the source and improves internal validity. **(2) Multi-level analysis**: The same data with before and after states supports analysis at the dataset and sample level. We introduce three delta (metrics) based diagnostics: the explained variance ratio (EVR), mean vertical distance (MVD) and above diagonal ratio (ADR), which respectively measure the noise level, discriminative strength, and baseline detection capability. We adjust MVD by EVR to obtain n-MVD, which characterizes the discriminative ability relative to the noise intensity. These metrics enable visual and quantitative tests of key MIA hypotheses. **(3) Transferability**: $\Delta$-MIA requires only data that remain unseen by the target model and does not require access to the original training corpus. This property lowers the barrier to evaluation across models and tasks.

Based on $\Delta$-MIA, we conduct a systematic re-evaluation of nine representative MIA methods. After removal of data distribution shift, many methods remain robust, but the performance of several methods such as DC-PDD (Zhang et al., 2024) and Con-ReCaLL (Wang et al., 2025) decreases sub-

stantially relative to original reports. This pattern indicates reliance on distributional shortcuts in prior evaluations and leads to inflated scores. The multi-level analysis enabled by $\Delta$-MIA provides deeper insight. Scatter visualizations of sample-level responses support precise qualitative assessment: tighter clusters indicate lower noise due to sample heterogeneity, and larger vertical distance above the diagonal signals stronger separation. For quantitative assessment, strong results of Min-K%++ (Zhang et al., 2025) arise from clear clustering and a high MVD score. We also observe a skewed distortion pattern in the scatter of ReCaLL (Xie et al., 2024), which explains the low true positive rate at low false positive rate. *All models, data, and code associated with this evaluation procedure will be made publicly available upon publication.*

## 2  $\Delta$-MIA EVALUATION FRAMEWORK

### 2.1  PROBLEM DEFINITION

A membership inference attack (MIA) on a target model $\mathcal{M}$ aims to determine if a given data sample $x$ belongs to the model's training dataset $\mathcal{D}$. The attack typically employs a scoring function $f(x; \mathcal{M})$ to quantify the likelihood of membership. This function is designed to capture signals of memorization, as models often exhibit different behaviors (e.g., lower loss) on data they were trained on. A prediction is then made by comparing the score against a threshold $\tau$:

$$\text{prediction}(x, \mathcal{M}) = \begin{cases} 1 \text{ (member)}, & \text{if } f(x; \mathcal{M}) \geq \tau, \\ 0 \text{ (non-member)}, & \text{if } f(x; \mathcal{M}) < \tau. \end{cases} \tag{1}$$

### 2.2  EVALUATION PROCEDURE

By following the popular setting (Duan et al., 2024; Shi et al., 2024), we implement our $\Delta$-MIA framework using the Pythia (Biderman et al., 2023) model suite (410M to 6.9B parameters) and its public training corpus, the Pile dataset (Gao et al., 2021). This setup provides the ground-truth knowledge of data membership required for a controlled and reliable evaluation.

**Constructing Pre- and Post-Exposure States.** Our procedure is designed to create model pairs that represent pre-exposure and post-exposure states, which forms the basis of the **self-contrast** principle.

First, we define a target dataset by sampling $1,500$ samples from the Pile test split (500 each from Pile-CC, Wikipedia, and Pubmed Abstracts). The base pre-trained Pythia models ($\mathcal{M}_{base}$) have not seen these instances; thus, they represent the pre-exposure state relative to this target data.

After that, to create the post-exposure state, we perform a two-stage, full-parameter fine-tuning process on each base model. (1) Data Exposure: The base model is fine-tuned on the $1,500$ target instances for one epoch. This step explicitly exposes the model to the data. (2) Stabilization: The model then continues training on $100,000$ different instances from the Pile training set. This second stage stabilizes the model and mitigates potential overfitting to the small target dataset. This two-stage process is critical, as it avoids evaluating the model immediately after exposure to the target data, which would yield artificially inflated MIA performance due to short-term, transient effects of recent training. Such inflated results are unfair and misaligned with pre-training MIA setups.

The resulting fine-tuned model is denoted as Pythia-post ($\mathcal{M}_{post}$). This procedure yields a model pair ($\mathcal{M}_{base}, \mathcal{M}_{post}$) for evaluation. An MIA method is then assessed by comparing the MIA scores it generates for the $1,500$ target instances from both models. This self-contrast design ensures that any observed difference in MIA scores is attributable to the membership signal rather than data distribution shifts.

**Auxiliary Data for Traditional Setups.** To ensure compatibility with MIA methods that require additional member and non-member sets, we also provide a small auxiliary dataset. We collect 60 instances from the Pile training split (seen by both $\mathcal{M}_{base}$ and $\mathcal{M}_{post}$) to serve as consistent members, and 60 instances from the validation split (unseen by both models) to serve as consistent non-members.

## 2.3 METRIC DESIGN

**Score Preprocessing.** To enable a fair, sample-level comparison across different MIA methods, we first preprocess the raw MIA scores obtained from $\mathcal{M}_{base}$ (non-member scores) and $\mathcal{M}_{post}$ (member scores). This process involves two key steps: (1) Outlier Removal, where we compute the 1st and 99th percentiles of all scores combined. A score pair for a given sample is retained only if both its non-member and member scores fall within this range, filtering extreme values without obscuring meaningful patterns. (2) Min-Max Normalization, where we scale the filtered scores to a common $[0, 1]$ interval using the global minimum and maximum from the combined filtered data. This ensures comparability across different MIA methods while preserving the relative distances between a sample's member and non-member scores.

**Sample-Level Diagnostic Metrics.** We introduce four metrics for a multi-faceted, sample-level analysis. We first establish two metrics, ADR and MVD, to measure an attack's fundamental detection ability and discrimination strength. We then introduce EVR and n-MVD to further quantify the impact of confounding noise and assess the signal-to-noise efficacy of the method. Let $(s_i^{\text{base (non-member)}}, s_i^{\text{post (member)}})$ denote the preprocessed score pair for the $i$-th of $K$ filtered samples.

**Above-Diagonal Ratio (ADR).** This metric measures the fundamental detection capability of an MIA method. It calculates the proportion of samples for which the post-exposure score is higher than the pre-exposure score. ADR is defined as:

$$\text{ADR} = \frac{1}{K} \sum_{i=1}^{K} \mathbb{I}(s_i^{\text{post (member)}} > s_i^{\text{base (non-member)}}), \tag{2}$$

where $\mathbb{I}(\cdot)$ is the indicator function. ADR should serve as the fundamental threshold for evaluating MIA effectiveness: failing to achieve a high ADR (e.g., above 90%) results in excessive false positives, which undermines confidence in the method's validity.

**Mean Vertical Distance (MVD).** While ADR captures the rate of correct detection, MVD quantifies the average strength of this detection. It measures the mean difference between post-exposure and pre-exposure scores. MVD is defined as:

$$\text{MVD} = \frac{1}{K} \sum_{i=1}^{K} (s_i^{\text{post (member)}} - s_i^{\text{base (non-member)}}). \tag{3}$$

A larger MVD indicates a stronger and more confident discrimination between member and non-member states.

**Explained Variance Ratio (EVR).** To quantify the influence of confounding noise, we introduce the EVR. This metric assesses the distribution of score pairs, which is often affected by inherent sample properties like text complexity. Such properties can create a strong linear correlation between pre- and post-exposure scores, obscuring the true membership signal. A robust MIA method should suppress this noise, causing the score pairs to form clusters rather than a linear trend.

The EVR is calculated using principal component analysis (PCA) (Maćkiewicz & Ratajczak, 1993) on the preprocessed score pairs. Let the score pairs form a data matrix $\mathbf{X} \in \mathbb{R}^{K \times 2}$, where each row is $\mathbf{X}_i = [s_i^{\text{base (non-member)}}, s_i^{\text{post (member)}}]$. After mean-centering the data to create $\hat{\mathbf{X}}$, we compute the covariance matrix $\Sigma = \frac{1}{K-1} \hat{\mathbf{X}}^T \hat{\mathbf{X}}$. By performing an eigenvalue decomposition on $\Sigma$, we obtain two eigenvalues, $\lambda_1 \geq \lambda_2$. The EVR is then defined as the proportion of variance explained by the first principal component:

$$\text{EVR} = \frac{\lambda_1}{\lambda_1 + \lambda_2}. \tag{4}$$

A high EVR (approaching 1) indicates that score variance is dominated by a single linear trend, suggesting that noise overshadows the membership signal. Conversely, a low EVR suggests the scores are more clustered, indicating that the membership signal is strong relative to confounding noise.

**Noise-Normalized MVD (n-MVD).** While MVD quantifies the raw discrimination strength and EVR assesses the level of confounding noise, neither metric alone captures the interplay between signal and noise. To provide a more holistic measure of an MIA method's efficacy, we introduce the

noise-normalized MVD (n-MVD). This metric evaluates the discrimination strength in proportion to the measured noise, effectively acting as a signal-to-noise ratio. It is defined as:

$$\text{n-MVD} = \frac{\text{MVD}}{\text{EVR}}. \tag{5}$$

A higher n-MVD score signifies a more robust and reliable method, as it indicates a strong membership signal (high MVD) that is clearly distinguishable from the underlying noise (low EVR). This allows for a more nuanced comparison between different MIA methods.

**Dataset-Level Metrics.** In addition to our sample-level diagnostics, we evaluate overall MIA performance using standard dataset-level metrics to ensure comparability with previous work (Carlini et al., 2021; Mireshghallah et al., 2022; Shi et al., 2024). We adopt two widely-used metrics: the Area Under the Receiver Operating Characteristic Curve (AUC) and the True Positive Rate at low False Positive Rates (TPR@low FPR).

We present the detailed dataset-level and sample-level analyses in Sec. 4.1 and Sec. 4.2, respectively.

## 3 EXPERIMENTAL SETTINGS

To facilitate consistent evaluation across methods, we standardize the interpretation of MIA scores. For attacks, a higher score indicates a greater likelihood that a sample is a member of training data. This standardization is a post-processing step for evaluation and does not alter the intrinsic mechanism of any method. Our evaluation includes the following nine representative MIA methods (for detailed hyperparameter settings, please refer to Appendix B.2):

1. **Loss** (Yeom et al., 2018). The score is the log-likelihood (LL) of the sample under the target model $\mathcal{M}$: $f(\mathbf{x}; \mathcal{M}) = \text{LL}(x; \mathcal{M})$
2. **Ref** (Carlini et al., 2020). This method calibrates the Loss score by subtracting the log-likelihood from a reference model ($\mathcal{M}_{\text{ref}}$), which accounts for the intrinsic predictability of the sample: $f(x; \mathcal{M}) = \text{LL}(x; \mathcal{M}) - \text{LL}(x; \mathcal{M}_{\text{ref}})$
3. **Zlib** (Carlini et al., 2020). The score is the Loss score normalized by the compressed size of the sample via the zlib algorithm, which serves as a proxy for sample complexity: $f(x; \mathcal{M}) = \text{LL}(x; \mathcal{M})/\text{zlib}(x)$
4. **Neighborhood** (Mattern et al., 2023). The score is the difference between the log-likelihood of the target sample and the average log-likelihood of its syntactically perturbed "neighbor" samples: $f(x; \mathcal{M}) = \text{LL}(x; \mathcal{M}) - \frac{1}{n} \sum_{i=1}^{n} \text{LL}(x_i'; \mathcal{M}_{\text{ref}})$
5. **Min-K%** (Shi et al., 2024). The score is the average log-likelihood of the tokens with the lowest K% predicted probabilities within the sample: $f(x; \mathcal{M}) = (1/|\text{min-}k(x)|) \sum_{x_i \in \text{min-}k(x)} \log p(x_i|x_{<i})$
6. **Min-K%++** (Zhang et al., 2025). This method enhances Min-K% by applying z-score normalization to the log-probability of each token before averaging: $f(x; \mathcal{M}) = (1/|\text{min-}k(x)|) \sum_{x_t \in \text{min-}k(x)} (\log p(x_t|x_{<t}) - \mu_{x_{<t}})/\sigma_{x_{<t}}$
7. **DC-PDD** (Zhang et al., 2024). The score measures the divergence between the model predictive distribution and an estimated token frequency distribution from an external corpus $D'$: $f(x; \mathcal{M}) = (1/|x|) \sum_{x_i \in x} p(x_i; \mathcal{M}) \log p(x_i; D')$
8. **ReCaLL** (Xie et al., 2024). The score is the ratio of the log-likelihood of the sample conditioned on a non-member prefix ($P_{\text{non-member}}$) to its unconditional log-likelihood: $f(x; \mathcal{M}) = \text{LL}(x|P_{\text{non-member}}; \mathcal{M})/\text{LL}(x; \mathcal{M})$
9. **Con-ReCaLL** (Wang et al., 2025). This method extends ReCaLL by incorporating a contrastive term based on the log-likelihood conditioned on a member prefix ($P_{\text{member}}$): $f(x; \mathcal{M}) = (\text{LL}(x|P_{\text{non-member}}; \mathcal{M}) - \gamma \cdot \text{LL}(x|P_{\text{member}}; \mathcal{M}))/\text{LL}(x; \mathcal{M})$

## 4 MAIN RESULTS

### 4.1 DATASET-LEVEL ANALYSIS: MODEL SCALES AND DATA DOMAINS

We present the dataset-level evaluation of nine MIA methods on our $\Delta$-MIA benchmark. We analyze performance from two primary perspectives: across varying model scales (Table 1) and different data domains (Table 2). The evaluation relies on AUC and TPR@5%FPR as the core metrics.

Table 1: Dataset-level results on the Δ-MIA Benchmark. We report AUC and TPR at 5% FPR (T@5%F) for MIA methods evaluated on the Pythia model family (410M, 1B, 2.8B, and 6.9B). Higher is better. The best score in each column is **bold**; the second best is underlined.

| Method | Pythia-410M | | Pythia-1B | | Pythia-2.8B | | Pythia-6.9B | |
|---|---|---|---|---|---|---|---|---|
| | AUC | T@5%F | AUC | T@5%F | AUC | T@5%F | AUC | T@5%F |
| Loss | 0.5622 | 0.0635 | 0.5853 | 0.0693 | 0.6348 | 0.0642 | 0.6582 | 0.0729 |
| Zlib | 0.5487 | 0.0851 | 0.5671 | 0.0967 | 0.6032 | 0.0952 | 0.6257 | 0.1176 |
| Neighborhood | 0.5555 | 0.0642 | 0.5895 | 0.0772 | 0.6525 | 0.0909 | 0.6828 | 0.1392 |
| Con-ReCaLL | 0.5841 | 0.0779 | 0.6014 | 0.1097 | 0.5792 | 0.0895 | 0.5957 | 0.1219 |
| DC-PDD | 0.5290 | 0.0469 | 0.5420 | 0.0527 | 0.6575 | 0.0996 | 0.6738 | 0.1133 |
| Min-K% | 0.5856 | 0.0758 | 0.6039 | 0.0765 | 0.6811 | 0.0887 | 0.6935 | 0.0952 |
| ReCaLL | **0.6782** | 0.0974 | **0.7257** | 0.1154 | 0.7515 | 0.1032 | 0.7343 | 0.1068 |
| Ref | 0.6781 | **0.1566** | 0.7020 | **0.1746** | 0.7530 | **0.2063** | 0.7620 | **0.2453** |
| Min-K%++ | 0.6583 | 0.0830 | 0.6771 | 0.0952 | **0.8327** | 0.1797 | **0.8054** | 0.1890 |

Table 2: Dataset-level results on the Δ-MIA Benchmark for the Pythia-2.8B model across three domains (Pile-CC, PubMed Abstracts, Wikipedia (en)). We report AUC and TPR at 5% FPR (T@5%F); higher is better. The best score in each column is **bold**; the second best is underlined.

| Method | Pile-CC | | PubMed | | Wikipedia | |
|---|---|---|---|---|---|---|
| | AUC | T@5%F | AUC | T@5%F | AUC | T@5%F |
| Loss | 0.6589 | 0.0713 | 0.6745 | 0.1487 | 0.6395 | 0.0837 |
| Zlib | 0.6029 | 0.0943 | 0.5954 | 0.1242 | 0.6531 | 0.0813 |
| Neighborhood | 0.6699 | 0.0776 | 0.6761 | 0.1629 | 0.6157 | 0.0383 |
| Con-ReCaLL | 0.4849 | 0.0629 | 0.6983 | 0.2383 | 0.5343 | 0.0813 |
| DC-PDD | 0.6656 | 0.1321 | 0.6835 | 0.1446 | 0.6767 | 0.0718 |
| Min-K% | 0.7003 | 0.1572 | 0.7276 | 0.2200 | 0.6858 | 0.1005 |
| ReCaLL | 0.8353 | 0.2600 | 0.8750 | 0.3829 | 0.7142 | 0.1053 |
| Ref | 0.7594 | 0.1803 | 0.7817 | 0.3035 | 0.7287 | **0.1675** |
| Min-K%++ | **0.8376** | **0.2935** | **0.8858** | **0.5886** | **0.7971** | 0.1411 |

**Analysis Across Model Scales.** We first investigate how the performance of MIA methods evolves as the target model parameter count increases. This analysis reveals key trends in model vulnerability and method scalability, leading to three primary findings. (1) Larger models are consistently more susceptible to MIAs. They tend to retain more pronounced membership signals, a trend that is particularly visible in the TPR@5%FPR metric. (2) A clear performance hierarchy emerges among the methods. The reference-based method, Ref, establishes a strong performance ceiling. Among reference-free approaches, Min-K%++ and ReCaLL are the leaders, though their performance scales differently: the effectiveness of Min-K%++ grows with model size, whereas the performance of Re-CaLL plateaus and exhibits a notable discrepancy between its high AUC and lower TPR@5%FPR. (3) Methods that potentially rely on distributional shortcuts underperform significantly in our bias-free setting. For instance, the previously reported effectiveness of Con-ReCaLL is neutralized without the distribution shifts present in other benchmarks. Similarly, DC-PDD struggles with the practical challenges of token frequency estimation in modern LLMs.

**Analysis Across Data Domains.** We examine how data characteristics influence attack efficacy by evaluating methods across distinct domains. The results reveal that MIA efficacy is highly sensitive to the data domain. As shown in Table 2, performance varies considerably across data sources. The effectiveness of certain methods, such as ReCaLL, significantly improves when using in-domain data for calibration, highlighting this dependency. Further details for all model scales are provided in Appendix C.

In summary, our analysis confirms that MIA vulnerability is critically dependent on both model scale and data domain. Our framework provides a more accurate assessment by establishing a clear performance hierarchy and revealing the failures of methods that rely on statistical shortcuts instead of genuine membership signals.

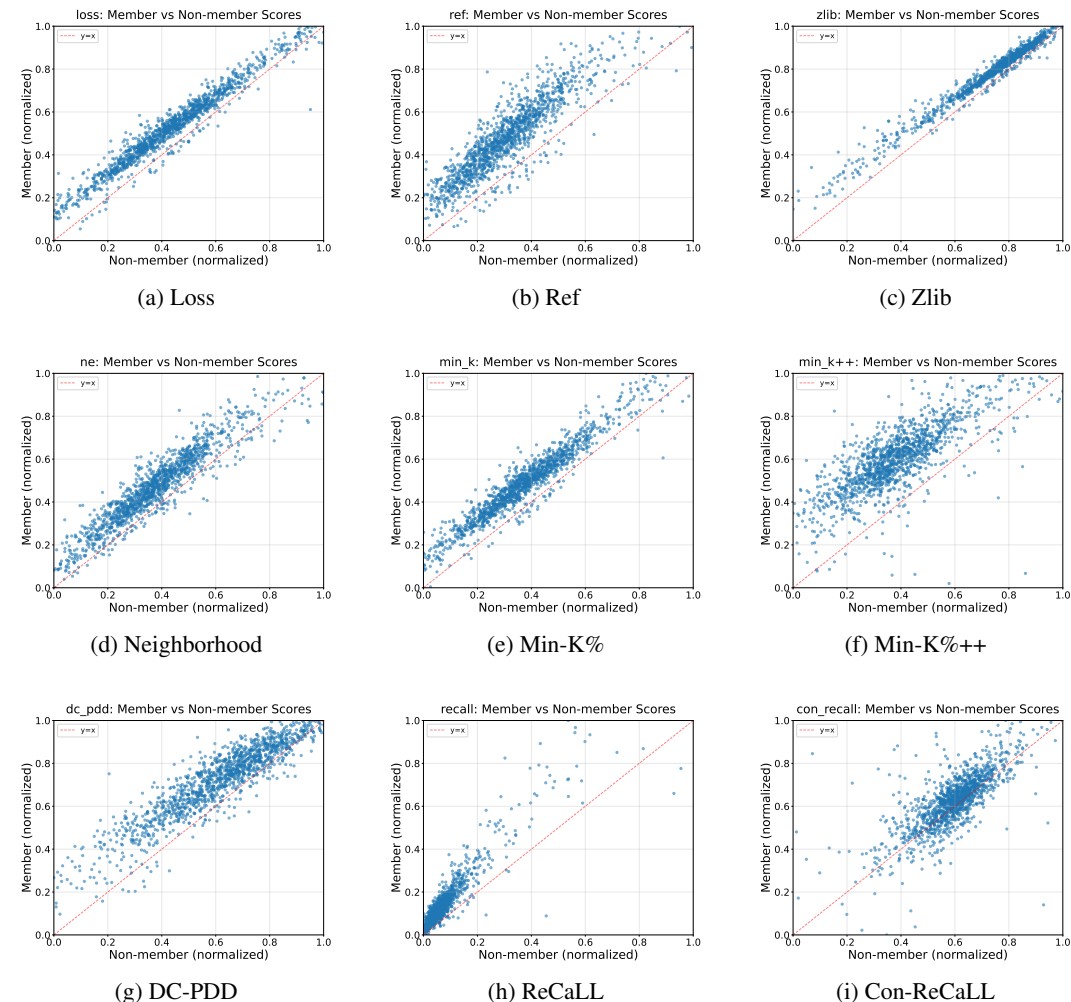

Figure 2: Sample-level scatter plots for Pythia-2.8B across 9 attack methods. Each subfigure shows the normalized member score ($y$-axis) versus the corresponding non-member score ($x$-axis), with the identity line $y = x$ for reference. Points above the line indicate correct membership reflection.

## 4.2 SAMPLE-LEVEL ANALYSIS: QUALITATIVE AND QUANTITATIVE DIAGNOSTICS

A fundamental limitation of existing MIA benchmarks is their reliance on an observational paradigm, which can confound the analysis of attack performance. Our $\Delta$-MIA framework addresses this by enabling a self-contrast evaluation. For each sample, we generate a pair of MIA scores corresponding to its pre-exposure (non-member) and post-exposure (member) states. Because the underlying data for both states is identical, distribution shift is eliminated as a confounding variable. Any observed difference in scores can therefore be attributed directly to the effects of model training, allowing for a fine-grained and interpretable analysis of the pure membership signal.

**Qualitative Analysis via Scatter Plots.** As shown in Fig. 2, we visualize the normalized score pairs in scatter plots to analyze the discriminative behavior of each MIA method. In these plots, non-member scores are on the x-axis and member scores are on the y-axis, and the diagonal line $y = x$ serves as a reference for correct inference. Our qualitative analysis reveals several key findings. (1) Most methods produce scatter plots with points concentrated above the diagonal, confirming their basic ability to capture membership signals. (2) The baseline Loss method exhibits a linear pattern just above the diagonal, indicating that its weak signal is heavily confounded by the noise from inherent sample difficulty. (3) In contrast, superior methods like Min-K%++ effectively suppress this noise, resulting in more clustered distributions and a larger vertical distance from the diagonal,

Table 3: Sample-level results on the Δ-MIA Benchmark. Performance of MIA methods across the Pythia family (410M, 1B, 2.8B, 6.9B) are evaluated with ADR and n-MVD. The highest value in each column is **bold**; the second highest is underlined.

| Method | Pythia-410M | | Pythia-1B | | Pythia-2.8B | | Pythia-6.9B | |
|---|---|---|---|---|---|---|---|---|
| | ADR(%) | n-MVD | ADR(%) | n-MVD | ADR(%) | n-MVD | ADR(%) | n-MVD |
| Loss | 94.7 | 0.041 | 94.2 | 0.056 | 95.6 | 0.090 | 91.6 | 0.107 |
| Zlib | 94.5 | 0.022 | 94.1 | 0.030 | 95.5 | 0.046 | 91.5 | 0.058 |
| Neighborhood | 86.4 | 0.034 | 90.7 | 0.055 | 92.6 | 0.095 | 92.0 | 0.112 |
| Con-ReCaLL | 79.1 | 0.047 | 84.1 | 0.054 | 68.7 | 0.036 | 74.3 | 0.029 |
| DC-PDD | 67.2 | 0.020 | 71.7 | 0.030 | 92.3 | 0.114 | 88.9 | 0.129 |
| Min-K% | 95.3 | 0.052 | 92.9 | 0.065 | 96.7 | 0.113 | 93.0 | 0.121 |
| ReCaLL | **97.8** | 0.057 | **98.4** | 0.062 | **97.9** | 0.068 | **96.2** | 0.059 |
| Ref | 94.7 | **0.107** | 94.3 | **0.117** | 95.8 | 0.177 | 91.7 | 0.187 |
| Min-K%++ | 92.6 | 0.093 | 90.3 | 0.110 | 96.6 | **0.246** | 92.8 | **0.241** |

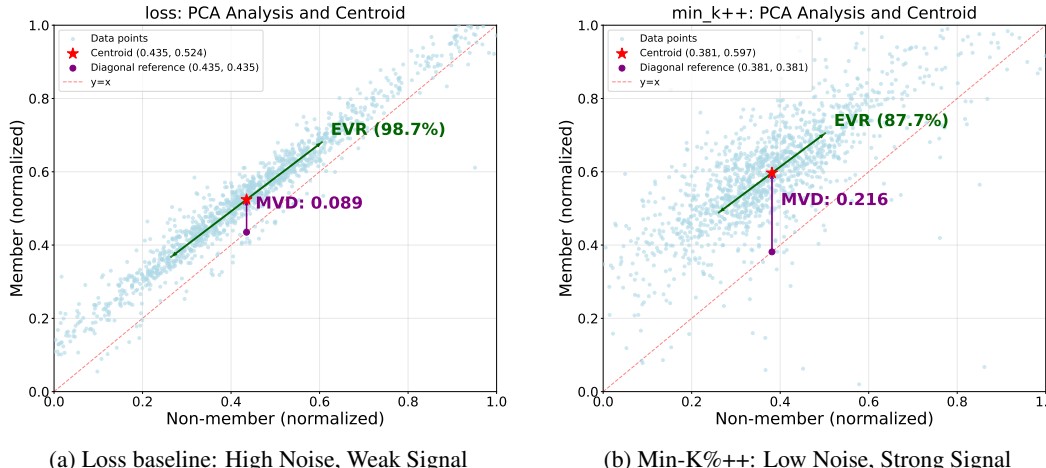

(a) Loss baseline: High Noise, Weak Signal      (b) Min-K%++: Low Noise, Strong Signal

Figure 3: PCA visualization of sample-level evaluations for MIA methods. Compared to **(a) the Loss baseline**, **(b) Min-K%++** demonstrates superior performance by effectively reducing sample-induced noise (lower EVR) while amplifying the membership signal (higher MVD). This validates the use of n-MVD as a signal-to-noise metric for MIA evaluation.

which signifies stronger discriminative power. (4) This fine-grained view also explains complex performance profiles. For example, the skewed score distribution of ReCaLL, characterized by extreme outliers, clarifies why it achieves a high AUC yet suffers from a low TPR at stringent thresholds. A detailed discussion is provided in Appendix D.

**Quantitative Analysis with Diagnostic Metrics.** The quantitative results from our diagnostic metrics, presented in Table 3, corroborate the qualitative findings. (1) The noise-normalized Mean Vertical Distance (n-MVD) tends to increase with model scale for most methods, confirming that larger models leave more distinct membership signals. (2) Methods exhibit distinct strengths: ReCaLL consistently achieves the highest Above-Diagonal Ratio (ADR), demonstrating a robust capability for basic detection. In contrast, Min-K%++ excels in n-MVD, particularly on larger models (e.g., 0.246 for Pythia-2.8B), indicating its superior ability to create a strong separation between memberand non-member scores.

**Further Discussion.** Fig. 3 visually decomposes the performance of a MIA method into noise intensity, represented by EVR, and discrimination strength, captured by MVD. This intuitive representation validates the formulation of n-MVD as a signal-to-noise metric for robust MIA evaluation.

In summary, our sample-level analysis provides crucial insights unattainable through aggregate metrics alone. (1) It reveals the underlying mechanisms of MIA methods, demonstrating that superior performance stems from suppressing noise related to sample difficulty, a factor that limits sim-

pler baselines like Loss. (2) This fine-grained perspective explains complex behaviors, such as the performance discrepancies in methods like ReCaLL, while our diagnostic metrics quantitatively distinguish between reliable detection (high ADR) and strong discriminative power (high n-MVD).

## 5 RELATED WORK

**Membership Inference Attacks (MIAs) in LLMs.** MIAs aim to compromise data privacy by identifying if a sample was in a model's training set (Shokri et al., 2017). For LLMs, these attacks largely fall into two categories (Wu & Cao, 2025). Target-model-based attacks, the most common approach, derive signals directly from the target model's outputs. The core assumption is that models respond differently to data they have memorized. Methods range from using simple metrics like model loss (Yeom et al., 2018) to more sophisticated signals, such as the likelihoods of the least probable tokens (e.g., Min-K% (Shi et al., 2024) and Min-K%++ (Zhang et al., 2025)) or score changes on perturbed sample neighbors (Mattern et al., 2023). Reference-model-based attacks calibrate the target model's score using an external model to account for a sample's intrinsic complexity, thus isolating the membership signal (Carlini et al., 2020; Mireshghallah et al., 2022; Fu et al., 2024). While effective in theory, this approach is often impractical for modern LLMs, as many are proprietary, and obtaining a suitably matched reference model is nearly impossible. Using a mismatched reference introduces its own confounding biases. A central challenge for both categories is that their reported effectiveness is heavily skewed by the evaluation setups used. The impracticality of reference models and the ambiguity of results from target-model-based attacks highlight an urgent need for a new evaluation paradigm. To mitigate these issues, our $\Delta$-MIA, provides such a evaluation framework, enabling a reliable assessment of these attacks without depending on impractical assumptions.

**Evaluation of Membership Inference Attacks.** Current MIA evaluation for LLMs predominantly follows an observational paradigm, which assesses attacks by comparing model responses on two distinct datasets: a member set from the training data and a non-member set from an external source. This approach is fraught with systemic flaws. For example, WikiMIA (Shi et al., 2024) employs a temporal split to create its non-member set, inadvertently introducing a distribution shift that allows attacks to learn temporal shortcuts. MIMIR (Duan et al., 2024) attempts to mitigate this with random splits and n-gram overlap filtering, but lexical similarity is a poor proxy for semantic distribution and fails to prevent unreliable results (Meeus et al., 2025b). The fundamental limitation of the observational paradigm is its reliance on constructing a non-member set, which inevitably introduces confounding variables like temporal or domain shifts. These shortcuts lead to inflated and misleading performance metrics. Moreover, this paradigm is often limited to coarse, dataset-level analysis and requires access to the original training corpus, hampering both granular insight and transferability. To address these limitations, we propose a shift to an interventional paradigm, actualized in our $\Delta$-MIA framework. Instead of observing static sets, we "intervene" by fine-tuning the model on previously unseen data. By comparing the model's behavior on the exact same data before (non-member) and after (member) this intervention, $\Delta$-MIA performs controlled, self-contrastive evaluations. This design eliminates data distribution shifts, enabling a more reliable, fine-grained, and transferable evaluation of MIA methods.

## 6 CONCLUSION

Prevailing observational paradigms for MIA evaluation in LLMs suffer from critical flaws, including data distribution shift, coarse-grained analysis, and poor transferability, which undermine the validity of existing benchmarks. To resolve these issues, we introduced Delta-MIA ($\Delta$-MIA), an interventional framework grounded in a self-contrast principle. By measuring changes in a model's behavior on the same data before and after exposure, $\Delta$-MIA isolates genuine membership signals from confounding artifacts. This design enables a bias-free, multi-level analysis and enhances transferability by removing the dependency on the original training corpus. Our systematic re-evaluation using this framework exposed previously inflated performance scores for several methods and provided deeper, mechanistic insights into their underlying behaviors. Ultimately, $\Delta$-MIA provides a more reliable and rigorous foundation for assessing privacy risks, auditing model compliance, and advancing the development of trustworthy AI systems.

## ETHICS STATEMENT

This research is dedicated to advancing the safe and responsible development of large language models, contributing to a broader agenda of AI safety and governance (Gyevnar & Kasirzadeh, 2025). Our proposed framework, $\Delta$-MIA, is designed as a defensive tool to provide a more reliable methodology for auditing privacy risks. We acknowledge the inherent dual-use nature of research in membership inference, where evaluation tools could inadvertently inform the development of more potent attacks (Meeus et al., 2025a). To mitigate this risk and primarily empower defenders, we will release our code and evaluation suite, fostering transparency and enabling the community to build robust safeguards. Furthermore, as a condition for using the publicly released artifacts, we will require that any derivative work or application explicitly states its intended purpose to ensure alignment with ethical research goals. All experiments were conducted on publicly available models and datasets, reinforcing our commitment to reproducible and accountable science.

## REPRODICIBILITY STATEMENT

All code, experimental data, and analysis code to reproduce the findings of this paper are provided in the supplementary material and are available at the following anonymous repository: `https://anonymous.4open.science/r/Delta-MIA`.

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

# Appendix

## A   Validation of the Foundational Assumption for $\Delta$-MIA

The core foundational assumption of $\Delta$-MIA is that lightweight fine-tuning on a small target dataset does not significantly alter the overall inference behavior of a large language model. This assumption is critical to the framework's validity: only if the model's intrinsic behavior remains stable can the observed differences between pre-exposure and post-exposure scores be attributed to genuine membership signals rather than spurious changes caused by fine-tuning. To fully verify this assumption and address potential concerns about evaluation robustness, we conduct two complementary experiments, presented in Sections A.1 and A.2.

### A.1   Validation with a Distribution-Mismatched Dataset (C4)

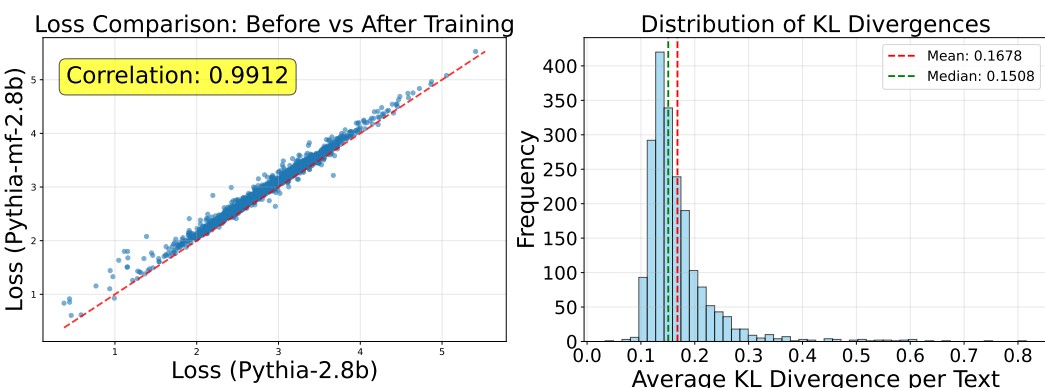

Figure 4: Left: A scatter plot of inference losses from Pythia-2.8B and Pythia-post-2.8B on data from C4. Right: A frequency histogram of the average Kullback-Leibler (KL) divergence per text sample.

We first validate the assumption using a dataset that is distributionally distinct from the target data. We perform an evaluation using the Pythia-2.8B base model and the fine-tuned counterpart, Pythia-post-2.8B. The evaluation uses $2,000$ text samples from the C4 dataset, which are unseen by both models and follow a different distribution from the target data. We employ two metrics for comparison. First, we record the inference loss for each sample from both models. Second, we compute the token-level Kullback-Leibler (KL) divergence between the vocabulary probability distributions of the two models to quantify the similarity of the outputs, using the base model as the reference distribution.

As shown in Fig. 4, the scatter plot on the left demonstrates a strong linear correlation between the inference losses of the base and fine-tuned models on identical unseen data. Furthermore, the histogram on the right shows that the average KL divergence per sample is concentrated near zero. The results illustrate that the model behavior remains stable after fine-tuning when evaluated on distribution-mismatched non-member data.

### A.2   Supplementary Validation with a Distribution-Matched Dataset (Pile Validation Split)

While Section A.1 verifies the assumption using a distribution-mismatched dataset, a critical consideration arises: if non-member data share the same distribution as the target data, could fine-tuning induce spurious changes in model behavior for these non-member samples? Such changes might lead to artificially inflated Mean Vertical Distance (MVD) values, undermining the reliability of $\Delta$-MIA 's evaluation results. To address this concern, we conduct a supplementary experiment using a distribution-matched non-member dataset.

We construct the evaluation dataset from the validation split of the Pile dataset, which shares the same distribution as the target data (sourced from the Pile test split). Specifically, we sample 500

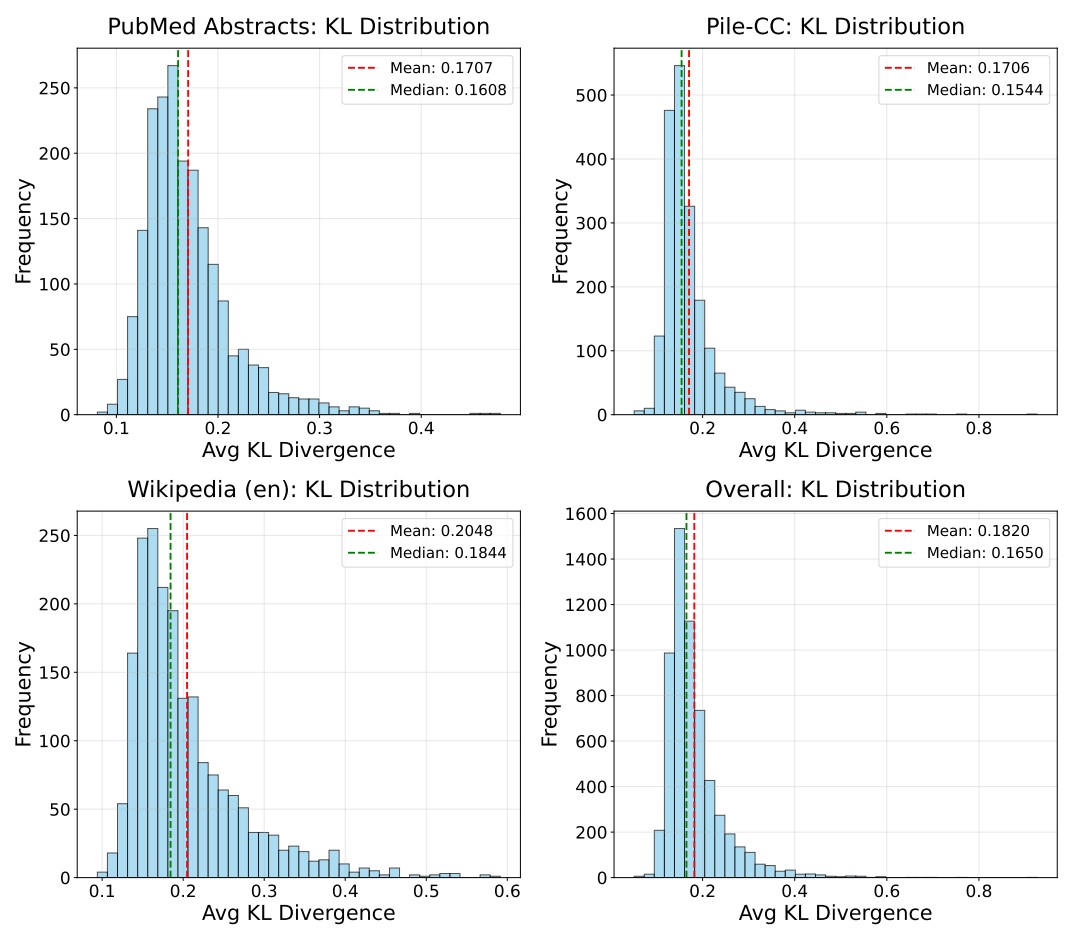

Figure 5: Distributions of average KL divergence between Pythia-2.8B and Pythia-post-2.8B on the distribution-matched non-member dataset (Pile validation split). From top-left to bottom-right: distributions for Pile-CC, PubMed Abstracts, Wikipedia, and the combined dataset. All distributions are concentrated near zero, confirming stable model behavior after fine-tuning.

samples from each of three domains (Pile-CC, PubMed Abstracts, and Wikipedia), forming a total of 1,500 samples. This dataset is unseen by both Pythia-2.8B and Pythia-post-2.8B, ensuring it serves as a pure non-member set for both model states while maintaining distributional alignment with the target data.

As shown in Fig. 5, the average KL divergence distributions for all three domains and the combined dataset are consistently concentrated near zero. Even when non-member data are distributionally matched to the target data, the fine-tuning process does not induce significant changes in the model's output distribution for these samples. The low KL divergence values confirm the robustness of $\Delta$-MIA that no spurious signals are introduced that could lead to false high MVD measurements.

These findings confirm that the model behavior remains stable after fine-tuning, thereby validating the core assumption that underpins the $\Delta$-MIA evaluation framework.

# B EXPERIMENTAL DETAILS

## B.1 FINE-TUNING CONFIGURATION

To ensure fair comparisons, we apply a consistent set of hyperparameters and optimization strategies across all Pythia model (Biderman et al., 2023) scales (410M, 1B, 2.8B, and 6.9B) during the fine-tuning process. The fine-tuning procedure involves two sequential stages. First, each base model

Table 4: Dataset-level results on the $\Delta$-MIA Benchmark for the Pythia-410M model across three domains (Pile-CC, PubMed Abstracts, Wikipedia (en)). We report AUC and TPR at 5% FPR (T@5%F); higher is better. The best score in each column is **bold**; the second best is underlined.

| Method | Pile-CC | | PubMed | | Wikipedia | |
|---|---|---|---|---|---|---|
| | AUC | T@5%F | AUC | T@5%F | AUC | T@5%F |
| Loss | 0.5605 | 0.0650 | 0.5845 | 0.0815 | 0.5737 | 0.0766 |
| Zlib | 0.5401 | 0.0608 | 0.5423 | 0.0896 | 0.5869 | 0.0933 |
| Neighborhood | 0.5555 | 0.0755 | 0.5709 | 0.0855 | 0.5524 | 0.0646 |
| Con-ReCaLL | 0.5416 | 0.0713 | 0.5822 | 0.0916 | 0.6348 | 0.1172 |
| DC-PDD | 0.5334 | 0.0545 | 0.5284 | 0.0428 | 0.5333 | 0.0311 |
| Min-K% | 0.5847 | 0.0901 | 0.6101 | 0.1120 | 0.5963 | 0.0909 |
| ReCaLL | **0.7491** | **0.2055** | **0.7847** | 0.2118 | **0.7181** | **0.1268** |
| Ref | 0.6463 | 0.1300 | 0.7016 | **0.2485** | 0.6968 | 0.1172 |
| Min-K%++ | 0.6768 | 0.1279 | 0.6792 | 0.1365 | 0.6263 | 0.0861 |

Table 5: Dataset-level results on the $\Delta$-MIA Benchmark for the Pythia-1B model across three domains (Pile-CC, PubMed Abstracts, Wikipedia (en)). We report AUC and TPR at 5% FPR (T@5%F); higher is better. The best score in each column is **bold**; the second best is underlined.

| Method | Pile-CC | | PubMed | | Wikipedia | |
|---|---|---|---|---|---|---|
| | AUC | T@5%F | AUC | T@5%F | AUC | T@5%F |
| Loss | 0.5915 | 0.0608 | 0.6116 | 0.1079 | 0.5985 | 0.0957 |
| Zlib | 0.5606 | 0.0755 | 0.5584 | 0.1079 | 0.6121 | 0.1124 |
| Neighborhood | 0.5899 | 0.0629 | 0.6077 | 0.1039 | 0.5810 | 0.0550 |
| Con-ReCaLL | 0.4570 | 0.0461 | 0.5610 | 0.0876 | 0.6725 | **0.1746** |
| DC-PDD | 0.5462 | 0.0755 | 0.5406 | 0.0530 | 0.5538 | 0.0263 |
| Min-K% | 0.6109 | 0.0881 | 0.6304 | 0.1426 | 0.6137 | 0.0981 |
| ReCaLL | **0.7671** | **0.1803** | **0.8229** | 0.2179 | **0.7188** | 0.1220 |
| Ref | 0.6884 | 0.1384 | 0.7216 | **0.2444** | 0.7072 | 0.1316 |
| Min-K%++ | 0.6842 | 0.1321 | 0.7061 | 0.2200 | 0.6547 | 0.0861 |

trains for one epoch on $1,500$ non-member samples sourced from the test split of the Pile dataset, covering three distinct domains. Second, the model undergoes further training for one additional epoch on $100,000$ samples from the training split of the Pile.

All models use the AdamW optimizer with a learning rate of $1 \times 10^{-5}$, a weight decay of $0.01$, and a gradient clipping threshold of $1.0$. For training efficiency, we leverage FP16 mixed precision and the DeepSpeed ZeRO-2 optimization suite, which incorporates gradient and optimizer state sharding.

A consistent effective batch size of 32 is maintained across all experiments. The 410M, 1B, and 2.8B models are trained on eight NVIDIA RTX 4090 GPUs (24GB), while the 6.9B model is trained on four NVIDIA A800 GPUs (80GB). To achieve the target batch size, the 410M and 1B models use a per-GPU batch size of 4, and the 6.9B model uses a per-GPU batch size of 8. These three models operate with a gradient accumulation step count of 1. The Pythia-2.8B model, however, requires a different setup due to memory constraints. It is trained with a per-GPU batch size of 1, and the gradient accumulation step count is increased to 4 to maintain the same effective batch size.

### B.2 DETAILS FOR HYPERPARAMETERS IN MIA EVALUATION

To ensure a fair evaluation across the nine MIA methods, we do not perform any hyperparameter tuning. Instead, we adopt the hyperparameter settings recommended in the original publication for each respective method. This approach is justified by the claims of hyperparameter robustness made by the authors in their work, which renders the selected configurations both reasonable and representative.

Table 6: Dataset-level results on the Δ-MIA Benchmark for the Pythia-6.9B model across three domains (Pile-CC, PubMed Abstracts, Wikipedia (en)). We report AUC and TPR at 5% FPR (T@5%F); higher is better. The best score in each column is **bold**; the second best is underlined.

| Method | Pile-CC | | PubMed | | Wikipedia | |
|---|---|---|---|---|---|---|
| | AUC | T@5%F | AUC | T@5%F | AUC | T@5%F |
| Loss | 0.6803 | 0.0943 | 0.6972 | 0.1853 | 0.6719 | 0.1196 |
| Zlib | 0.6214 | 0.1237 | 0.6119 | 0.1650 | 0.6944 | 0.1411 |
| Neighborhood | 0.6912 | 0.1279 | 0.7022 | 0.2098 | 0.6536 | 0.0670 |
| Con-ReCaLL | 0.5095 | 0.0650 | 0.4051 | 0.0407 | 0.6567 | 0.1746 |
| DC-PDD | 0.6798 | 0.1363 | 0.6987 | 0.1589 | 0.6986 | 0.0933 |
| Min-K% | 0.7129 | 0.1551 | 0.7350 | 0.2587 | 0.7046 | 0.1292 |
| ReCaLL | 0.8004 | 0.2117 | 0.8286 | 0.2566 | 0.7075 | 0.1196 |
| Ref | 0.7675 | 0.2348 | 0.7843 | 0.3707 | 0.7503 | 0.1435 |
| Min-K%++ | **0.8076** | **0.3082** | **0.8395** | **0.4847** | **0.7897** | **0.1746** |

The specific hyperparameters used in evaluation are as follows:

1. **Ref** (Carlini et al., 2020): Pythia-70M as reference model.
2. **Neighborhood** (Mattern et al., 2023): 10 neighbor samples generated per target sample.
3. **Min-K% & Min-K%++** (Shi et al., 2024; Zhang et al., 2025): Average (original or normalized) log-likelihood computed over the 20% of tokens with the lowest probabilities.
4. **DC-PDD** Zhang et al. (2024): Upper bound $a = 0.01$ for the cross-entropy between individual token probability distributions and frequency distributions.
5. **ReCaLL** (Xie et al., 2024): Prefix setting with num_shots $= 1$.
6. **Con-ReCaLL** (Wang et al., 2025): Prefix setting with num_shots $= 1$, and non-member prefix contrast coefficient $\gamma = 1$.

## C  DETAILS OF DOMAIN-SPECIFIC MIA PERFORMANCE

This section expands upon the analysis presented in Sec. 4.1 by providing supplementary, domain-specific results for the nine evaluated MIA methods. The Table 4, Table 5 and Table 6 detail the performance across three distinct data domains (Pile-CC, PubMed Abstracts, and Wikipedia) for Pythia models of three different scales.

## D  INTERPRETING SCATTER PLOTS FOR ROC METRICS

The geometry of a scatter plot, which maps normalized non-member scores to the x-axis and member scores to the y-axis, directly encodes the trade-off between the True Positive Rate (TPR) and the False Positive Rate (FPR). This section formalizes this connection to explain why ReCaLL underperforms Ref on the TPR@5%FPR metric, linking the mathematical definitions directly to the visual features in Fig. 6.

Let $s_{mem}$ and $s_{non-mem}$ denote the normalized scores for members and non-members, respectively. For a given decision threshold $\tau$, we define four quadrants in the score space:

- Top-Left, TL($\tau$): $\{s_{mem} \geq \tau \wedge s_{non-mem} < \tau\}$

- Top-Right, TR($\tau$): $\{s_{mem} \geq \tau \wedge s_{non-mem} \geq \tau\}$

- Bottom-Left, BL($\tau$): $\{s_{mem} < \tau \wedge s_{non-mem} < \tau\}$

- Bottom-Right, BR($\tau$): $\{s_{mem} < \tau \wedge s_{non-mem} \geq \tau\}$

The TPR and FPR are the fractions of points lying above the horizontal line $y = \tau$ and to the right of the vertical line $x = \tau$, respectively. To evaluate TPR@5%FPR, the threshold $\tau$ is set to the 95th percentile of the non-member scores, which by definition fixes the FPR at 5%. The TPR is then the

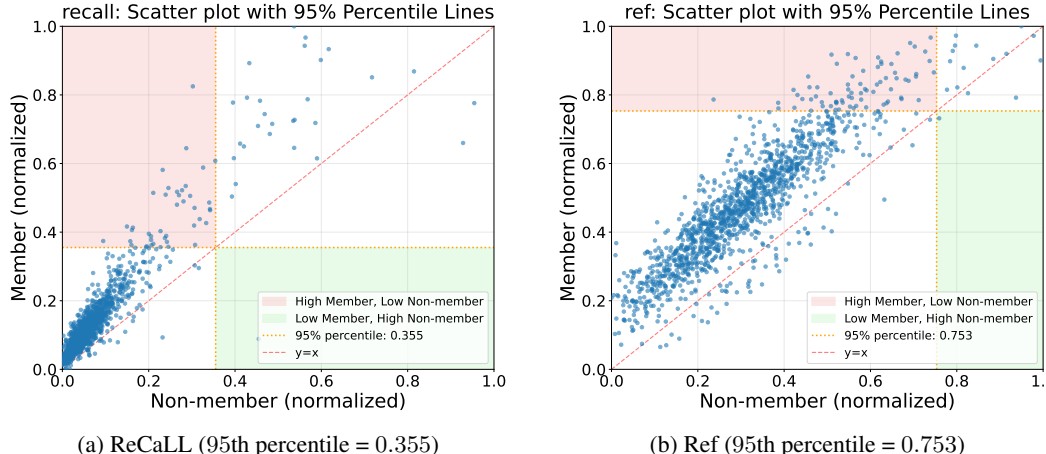

(a) ReCaLL (95th percentile = 0.355)   (b) Ref (95th percentile = 0.753)

Figure 6: Visualizing the impact of score distribution on TPR@5%FPR. (a) The scatter plot for ReCaLL exhibits a distribution skewed toward low non-member scores. This results in a low 95th percentile threshold ($\tau = 0.355$) and a sparsely populated top-left (TL) region, leading to a low TPR. (b) In contrast, the Ref method produces a more dispersed distribution, yielding a higher threshold ($\tau = 0.753$) and a dense TL region, which corresponds to a high TPR. The diagonal dashed line indicates a random-guess baseline. The red shaded area represents the TL($\tau$) quadrant, while the green area represents the BR($\tau$) quadrant.

proportion of member points above this line:

$$\text{TPR} = \frac{|\text{TL}(\tau)| + |\text{TR}(\tau)|}{N} \tag{6}$$

where N is the total number of members. The performance gap, TPR - FPR, simplifies to:

$$\text{TPR} - \text{FPR} = \frac{|\text{TL}(\tau)| - |\text{BR}(\tau)|}{N}. \tag{7}$$

This equation confirms that superior performance at a low FPR depends on maximizing the density of points in the top-left (red) region while minimizing points in the bottom-right (green) region.

The scatter plot for ReCaLL reveals a distribution heavily skewed to the left, indicating that most samples receive very low non-member scores. Consequently, the 95th percentile threshold $\tau$ required to achieve a 5% FPR is itself very low ($\tau = 0.355$). Although this stringent threshold effectively limits false positives, it also penalizes the true positive count. Because a significant portion of the member scores in the ReCaLL distribution are also concentrated in this low-score range, many fall below the line $y = \tau$. This results in a sparsely populated top-left region (TL($\tau$)) and, therefore, a low TPR. In contrast, the Ref method produces a more dispersed distribution, leading to a higher threshold ($\tau = 0.753$) that a larger fraction of member points can surpass, yielding a dense TL($\tau$) region and a correspondingly high TPR.

# E   SUPPLEMENTARY OBSERVATIONAL EVALUATION AND PARADIGM EQUIVALENCE VERIFICATION

To further assess the relationship between the interventional $\Delta$-MIA setup and the traditional observational formulation of membership inference, we conduct an additional evaluation using a post-exposure model (Pythia-post) as the target. This experiment serves two purposes: (1) verifying whether $\Delta$-MIA isolates the same underlying membership signal that observational MIAs aim to capture, and (2) examining the consistency of performance trends across evaluation paradigms. The results also highlight the inherent limitations of observational benchmarks in fully eliminating distributional confounders.

We conduct a standard observational MIA evaluation using the post-exposure Pythia models (410M, 1B, 2.8B, and 6.9B) as targets. The member set consists of the 1,500 samples used as exposure

Table 7: Dataset-level results of the supplementary observational evaluation on post-exposure models. We report AUC and TPR at 5% FPR (T@5%F) for MIA methods evaluated on the Pythia model family (410M, 1B, 2.8B, and 6.9B). Higher is better. The best score in each column is **bold**; the second best is underlined.

| Method | Pythia-410M | | Pythia-1B | | Pythia-2.8B | | Pythia-6.9B | |
|---|---|---|---|---|---|---|---|---|
| | AUC | T@5%F | AUC | T@5%F | AUC | T@5%F | AUC | T@5%F |
| Loss | 0.5735 | 0.0693 | 0.5981 | 0.0743 | 0.6894 | 0.1089 | 0.6860 | 0.1017 |
| Zlib | 0.5691 | 0.1025 | 0.5889 | 0.1227 | 0.6628 | 0.2020 | 0.6620 | 0.2063 |
| Neighborhood | 0.5186 | 0.0685 | 0.5434 | 0.0736 | 0.6406 | 0.1017 | 0.6325 | 0.1039 |
| Con-ReCaLL | 0.4457 | 0.0469 | 0.4548 | 0.0620 | 0.4624 | 0.0606 | 0.4654 | 0.0772 |
| DC-PDD | 0.6000 | 0.0866 | 0.6327 | 0.1248 | 0.7454 | 0.2439 | 0.7395 | 0.2150 |
| Min-K% | 0.5960 | 0.0830 | 0.6258 | 0.0895 | 0.7375 | 0.1537 | 0.7288 | 0.1645 |
| ReCaLL | 0.5603 | 0.0729 | 0.6076 | 0.0649 | 0.6997 | 0.1032 | 0.6972 | 0.0945 |
| Ref | 0.6480 | **0.1118** | 0.6806 | **0.1551** | 0.8189 | **0.3521** | 0.7795 | 0.3016 |
| Min-K%++ | **0.6569** | 0.1075 | **0.7114** | 0.1501 | **0.8524** | 0.3499 | **0.8415** | **0.3304** |

data in $\Delta$-MIA, while the non-member set is formed by another 1,500 samples drawn from the Pile validation split with matched domain composition (Pile-CC, PubMed Abstracts, and Wikipedia). All MIA methods are evaluated with the hyperparameter configurations of B.2, identical to those used in $\Delta$-MIA. We report AUC and TPR@5%FPR to facilitate direct comparison with the interventional results presented in Table 1; the corresponding observational results are shown in Table 7.

**Paradigm Equivalence Verification.** Across most methods, performance trends are highly consistent between the observational setting and the $\Delta$-MIA setup. Reference-based Ref and reference-free Min-K%++, the strongest performers in $\Delta$-MIA, remain the top methods under the observational protocol and exhibit similar scaling with model size. Baselines such as Loss and Zlib, as well as mid-tier approaches including Neighborhood and Min-K%, also show largely aligned relative performance. This consistency indicates that $\Delta$-MIA captures the same core membership signal targeted by traditional MIA evaluations. By holding the data distribution fixed and varying only exposure, $\Delta$-MIA provides a controlled environment for isolating genuine memorization effects, while remaining aligned with the conceptual objective of observational MIA. Moreover, the results suggest that $\Delta$-MIA's findings transfer to real-world settings where the primary concern is detecting true memorization rather than exploiting incidental distributional differences.

**Method-Specific Discrepancies.** Two methods exhibit notable differences across evaluation paradigms, arising from their intrinsic design:

1) ReCaLL. Its performance decreases substantially in the observational setting. ReCaLL depends heavily on the distribution of prefix data used in its conditional likelihood computation. In $\Delta$-MIA, where the same dataset serves as both pre- and post-exposure probes, prefix distributions are perfectly matched, mitigating this source of variation. Under the observational setup, even with distributionally aligned member/non-member sets, subtle differences in the impact of prefix data persist, degrading ReCaLL's performance. This is exacerbated by the absence of hyperparameter tuning, which we opted to forgo to ensure fairness across MIA methods.

2) DC-PDD. This method performs much better under the observational paradigm than under $\Delta$-MIA. The method estimates token frequency distributions from external corpora, making it sensitive to any distributional differences between member and non-member sets. Observational benchmarks, even when carefully constructed, cannot fully eliminate such subtle discrepancies. $\Delta$-MIA removes these shortcuts entirely, exposing the method's limited ability to capture true membership signals.

## F  THE USE OF LARGE LANGUAGE MODELS

The role of large language models (LLMs) in the preparation of this paper is strictly confined to improving the language, clarity, and readability of the manuscript. LLMs were not used for any substantive research tasks, including but not limited to generating hypotheses, performing data analysis, conducting literature reviews, or interpreting results.

