# OpenReview forum: "Delta-MIA: Measuring Membership Inference Attacks in Large Language Models via self-Contrast Framework"
_ICLR.cc/2026/Conference — Submitted to ICLR 2026_

### Official Review · Reviewer_nYoz · 2025-10-28

**Soundness:** 2
**Presentation:** 3
**Contribution:** 2
**Rating:** 4
**Confidence:** 5

**Summary:**

This paper introduces Delta-MIA (Δ-MIA), an interventional self-contrast framework for evaluating membership inference attacks in large language models. The work argues that existing MIA benchmarks are fundamentally limited by data distribution shift, coarse-grained analysis, and dependence on proprietary corpora, which confound evaluation and inflate performance metrics.

The key contribution is the interventional paradigm: instead of constructing separate member/non-member datasets, Δ-MIA compares a model’s responses before and after exposure to the same dataset. This self-contrast design eliminates distributional confounders and enables sample-level analysis through newly proposed diagnostics—ADR, MVD, EVR, and n-MVD—which quantify detection accuracy, discriminative strength, and noise sensitivity.

**Strengths:**

The research problem is clearly defined, and the paper's writing is good, making it easy for the reader to follow.
The paper introduces four well-motivated diagnostic metrics (ADR, MVD, EVR, n-MVD) that provide fine-grained insights into model behavior.
Also, the paper presents a solid empirical result from an experiment on their proposed evaluation method.
The authors commit to open-sourcing all code and evaluation data, and clearly articulate ethical safeguards to prevent misuse.

**Weaknesses:**

While the paper raises an important issue—the risk that current MIA benchmarks overestimate attack effectiveness due to distributional shortcuts—the motivation and the validation of Δ-MIA are not fully aligned.
The central claim is that Δ-MIA provides a faithful, bias-free framework for assessing membership inference in LLMs. However, the experiments mainly show that some existing MIA methods still yield non-trivial results under the Δ-MIA setting. This empirical observation alone does not demonstrate that Δ-MIA truly measures the same construct as “real-world” MIA evaluation.

Specifically, the paper does not theoretically or empirically establish equivalence between the Δ-MIA setting (before/after fine-tuning on a held-out dataset) and the conventional MIA problem (evaluating a pretrained model on candidate data with no distribution shift). Without such validation, it remains unclear whether the deltas measured by Δ-MIA genuinely reflect membership signals rather than overfitting signals of fine-tuning or optimization.

A stronger demonstration—either a theoretical argument showing that Δ-MIA preserves the same membership decision boundary as the standard setting, or an empirical study comparing Δ-MIA scores with ground-truth membership probabilities in a controlled environment—would significantly strengthen the paper’s central claim.

**Questions:**

1. What if the MIA method relies on the fine-tuning process? Can the D-MIA still be a good evaluation framework for MIA?  For example, the technique proposed in the paper "FINE-TUNING CAN HELP DETECT PRETRAINING DATA FROM LARGE LANGUAGE MODELS"

2. As I mentioned in the weakness part, can you show some evidence that if the MIA works under the Delta-MIA, which means it can evaluate a pretrained model on candidate data with no distribution shift. Because I still think fine-tuning cannot approximate pretraining, even if you add extra samples during fine-tuning to avoid overfitting. If you can persuade me that the MIAs working under Delta-MIA indicate a clear membership signal, I will raise the score.

3. Which subset did you use in MIMIR?  7-gram? 13-gram? It is essential because the MIA methods you showed in the paper all fail (close to random guessing) on 13-gram subsets of MIMIR (Wiki, arXiv, PubMed).

---

> ### Author Response · Authors · 2025-11-17
> **Part 1/2 of the Response to Reviewer nYoz**
>
> We sincerely thank you for the thoughtful comments and constructive suggestions. We have revised the paper accordingly, and all newly added analyses and results have been incorporated into the updated version. Below we summarize your concerns and provide point-by-point responses. Due to space limitations, we will respond in two parts.
>
> ### **(1) Concern that the fine-tuning setup in Δ-MIA does not faithfully simulate full pretraining**
>
> **Summary:**
> The reviewer argues that the fine-tuning used in Δ-MIA cannot replicate the behavior of a full pretraining run, and therefore questions whether the resulting MIA signals are representative. (from Question 2)
>
> **Response:**
> We appreciate the reviewer’s perspective. It is correct that our fine-tuning stage does not, and need not reproduce a full pretraining trajectory. The goal of Δ-MIA is not to achieve *absolute parity* with full-scale pretraining, but rather to create a **controlled, realistic, and analyzable training regime** for evaluation.
>
> There may be a misunderstanding regarding the intention of our design. We do not treat fine-tuning as a substitute for complete pretraining but as **one additional pretraining-like stage** positioned near the end of a long training trajectory. As discussed in MIMIR’s analysis of training-step effects (“Performance on benchmarks with more recently seen members is higher, but gradually decreases to a plateau for less recently seen members”), the primary challenge is to avoid evaluating models immediately after exposure when MIA performance would be artificially inflated.
>
> Our choice of \~100k samples (≈10^8 tokens) is informed directly by this observation: this amount of training is sufficient to move the model **past the extreme “just-seen” region** and into the stable “plateau”, where membership signals remain detectable but not excessively amplified. At the same time, we keep the learning rate and epoch count aligned with typical pretraining hyperparameters, ensuring that the optimization dynamics remain compatible with those of a pretraining stage.
>
> Importantly, Δ-MIA is designed to ensure **fair relative comparison** across MIA methods and models. Achieving absolute equivalence with full pretraining is unnecessary. Instead, what matters is (1) consistency of recency across all target samples, and (2) a stable regime where systematic upward shifts in signal strength due to shorter training affect **all methods equally** and therefore do not alter their relative ordering. Our two-stage setup satisfies exactly these requirements.
>
> ### **(2) Concern whether Δ-MIA can be used to evaluate fine-tuning-based MIA methods**
>
> **Summary:**
> The reviewer asks whether Δ-MIA is compatible with evaluating MIA methods that rely on *fine-tuning with non-members* to amplify membership signals. (from Question 1)
>
> **Response:**
> We appreciate the reviewer for this question. The method introduced in this paper is conceptually similar to ReCaLL, which tracks how perplexity changes when applying in-context learning with non-members, whereas this method measures perplexity (or another score function) changes when fine-tuning the model on non-members. Both methods rely on *perturbing model behavior via controlled exposure to non-member samples* and observing the resulting shift.
>
> Because Δ-MIA is designed as a **general benchmarking framework**, it can in principle accommodate this class of methods as well. However, incorporating fine-tuning-based attacks would require symmetric procedural adjustments that applying the fine-tuning perturbation to both pre-exposure and post-exposure models to ensure that the contrast isolates genuine membership signals rather than artifacts of the perturbation process.
>
> It is also worth noting that *fine-tuning-based MIA is not a standalone attack paradigm*—it ultimately depends on a **score function** whose shift under non-member fine-tuning is used to infer membership. From this perspective, Δ-MIA remains fully relevant: by evaluating and identifying **stronger, more stable score functions** through Δ-MIA, we can directly strengthen fine-tuning-based attacks, regardless of the underlying perturbation mechanism.
>
> ### **(3) Clarification on data selection details (e.g., which MIMIR subset was used)**
>
> **Summary:**
> The reviewer asks for clarification regarding which MIMIR subset was used for constructing non-member data. (from Question 3)
>
> **Response:**
> We suppose there might be a misunderstanding here. We would like to clarify that our benchmark does not rely on any MIMIR subsets. MIMIR’s n-gram filtering is designed for comparing *member vs. non-member* distributions, whereas in our setting we only need **non-member data** (and its duality as member to post-exposure models), so the n-gram criterion is not applicable.
>
> For our experiments, we directly sample from the **Pile** dataset following the criteria described in Section 2.2, without adopting MIMIR’s data splits or n-gram thresholds.

---

> > ### Author Response · Authors · 2025-11-17
> > **Part 2/2 of the Response to Reviewer nYoz**
> >
> > ### **(4) Concern about paradigm equivalence between Δ-MIA and conventional MIA setup**
> >
> > **Summary:**
> > The reviewer observes that several MIA methods exhibit non-trivial performance under Δ-MIA, yet behave close to random guessing under train/test-split observational benchmarks such as MIMIR. This raises concerns that Δ-MIA might be capturing overfitting artifacts from fine-tuning rather than genuine membership signals, and suggests the need for theoretical or empirical justification supporting the equivalence of the two evaluation paradigms. (from Weakness 1,2,3 and Question 2,3)
> >
> > **Response:**
> > We appreciate this thoughtful question and provide clarification across several aspects.
> >
> > ***
> >
> > #### **(a) Why Δ-MIA yields non-trivial performance**
> >
> > The higher performance observed under Δ-MIA is expected. While the \~100k-sample second-stage training moves the model beyond the “just-seen” region, MIMIR’s analysis shows that membership signals continue to decrease slowly as training proceeds. Because Δ-MIA necessarily operates with a shorter continuation of training than trillion-token pretraining, the resulting MIA signals are somewhat stronger. Importantly, this systematic upward shift affects **all** MIA methods equally, leaving **relative comparisons** across methods and models fully fair and interpretable.
> >
> > ***
> >
> > #### **(b) Why train/test-split benchmarks often yield near-random results**
> >
> > Many prior works have attributed the near-random behavior to the scale and nature of LLM pretraining—near-1-epoch passes over enormous corpora make the majority of samples effectively “far in the past” and therefore hard to distinguish.
> >
> > Recent evidence further shows that **LLMs exhibit forgetting during pretraining**\[1]. If a model has already forgotten certain training samples, then poor MIA performance on these samples is natural, even if they are labeled as “members.” This implies that train/test-split benchmarks may **underestimate** the true strength of MIA methods because they include a substantial number of “forgotten members” for which no attack could reasonably succeed.
> >
> > Crucially, this is not a failure of MIA; it reflects the mismatch between the benchmark design and realistic privacy concerns. The objective of MIA is to detect memorized (and potentially privacy-sensitive) content; requiring attacks to recover forgotten examples adds artificial difficulty and drifts away from real-world safety considerations.
> >
> > Notably, newer MIA methods such as Min-K%++ have already achieved **AUC ≈ 0.6** on Pythia-12B even under the difficult MIMIR setting, demonstrating that “all methods fail” is not an inherent property of MIA.
> >
> > ***
> >
> > #### **(c) Empirical verification of paradigm equivalence**
> >
> > Following the reviewer wA5G’s suggestion, we constructed a 1,500-sample non-member set drawn from the same domains of the Pile validation set and conducted an additional **observational evaluation** on post-exposure models (details in Appendix E of the updated version).
> >
> > The results show that **most MIA methods (e.g., Min-K%++, Ref, Loss)** exhibit performance patterns that closely match their Δ-MIA behavior. This strong consistency across paradigms indicates that the Δ-MIA signal reflects genuine membership effects.
> >
> > For the few methods whose behaviors diverge (e.g., ReCaLL, DC-PDD), our analysis attributes the differences to **method-specific sensitivities,** such as dependency on prefix distribution or susceptibility to distributional shortcuts, rather than to a fundamental discrepancy between the two evaluation paradigms. We discuss these cases in detail in Appendix E.
> >
> > These findings empirically support the alignment between Δ-MIA and traditional MIA objectives.
> >
> > ***
> >
> > #### **(d) On distinguishing “membership signal” from “overfitting signal”**
> >
> > We do not view these as opposing concepts. Membership signal is defined as evidence exhibited by a model **after** training on a sample. Overfitting signal is simply the (temporarily stronger) evidence shown **immediately after** exposure.
> >
> > Δ-MIA intentionally avoids the extreme “just-seen” regime by using \~100k continuation steps. What remains is stable membership signal, not noise due to overfitting.
> >
> > ***
> >
> > #### **(e) Why Δ-MIA is necessary in the first place**
> >
> > Current observational benchmarks cannot perfectly eliminate distributional shift. Even MIMIR’s n-gram alignment operates only at the lexical level and cannot fully control semantic-level discrepancies. Besides, even under the “without thresholding” setting, the GitHub subset still shows both high MIA performance and high n-gram similarity, contradicting the benchmark’s underlying assumptions, remaining unaccountable.
> >
> > This motivates the design of Δ-MIA: by using **self-contrast** between pre- and post-exposure models on the **same data**, we eliminate distribution shift at its root and obtain a cleaner view of true membership effects.
> >
> > \[1] Exploring Forgetting in Large Language Model Pre-Training (ACL 2025)

---

### Official Review · Reviewer_1iHS · 2025-10-30

**Soundness:** 2
**Presentation:** 3
**Contribution:** 2
**Rating:** 4
**Confidence:** 4

**Summary:**

This paper introduces Delta-MIA, a comprehensive framework for measuring membership inference attacks (MIAs) on fine-tuned large language models (LLMs). The authors propose a set of novel metrics (Above-Diagonal Ratio, Mean Vertical Distance, Explained Variance Ratio, and Noise-Normalized MVD) that provide a more nuanced evaluation of MIA effectiveness beyond traditional dataset-level metrics like AUC. The framework validates the core assumption that fine-tuning doesn't significantly change model behavior by comparing pre- and post-fine-tuning model outputs. The paper evaluates nine representative MIA methods across three domains (Pile-CC, PubMed Abstracts, Wikipedia) using the proposed metrics, providing insights into the relative effectiveness of different approaches.

**Strengths:**

S1: The paper introduces a comprehensive evaluation framework that goes beyond standard metrics (AUC, TPR@FPR) to provide deeper insights into MIA effectiveness through multiple complementary metrics.

S2: The proposed metrics (ADR, MVD, EVR, n-MVD) are theoretically well-motivated and address key limitations of existing evaluation approaches, particularly the lack of nuance in current evaluation practices.

S3: The framework successfully validates the core assumption that fine-tuning doesn't significantly alter model behavior, which is crucial for the validity of the Delta-MIA evaluation framework.

S4: The paper provides a clear benchmark for evaluating MIA methods across different domains, which will be valuable for future research in this area.

S5: The empirical results (Table 4) clearly demonstrate the utility of the proposed metrics, showing that methods like ReCall and Min-K%++ consistently perform well across different domains.

**Weaknesses:**

W1: The paper lacks sufficient comparison with existing MIA evaluation frameworks, making it difficult to fully appreciate the novelty of the proposed metrics.

W2: The evaluation is limited to Pythia models across only three domains, which limits the generalizability of the findings to other LLM architectures and datasets.

W3: The paper doesn't adequately address the practical implications of the proposed metrics for real-world privacy risk assessment and defense mechanisms.

W4: The theoretical justification for the proposed metrics could be strengthened with more detailed mathematical analysis and comparison to related work.

W5: The paper doesn't explore the relationship between the proposed metrics and the actual privacy risk in fine-tuned LLMs, which is the ultimate concern for the field.

**Questions:**

Q1: Could you provide a more detailed comparison between your proposed metrics and existing evaluation metrics (AUC, TPR@FPR) to better demonstrate the added value of your framework?

Q2: How would the proposed metrics perform when evaluated against different LLM architectures (e.g., GPT, LLaMA, Qwen) beyond the Pythia models used in your experiments?

Q3: Could you explore the relationship between your proposed metrics (especially n-MVD) and actual privacy risk, potentially by comparing with real-world privacy leakage measurements?

Q4: How would the Delta-MIA framework be adapted to evaluate MIAs on different types of fine-tuned models (e.g., instruction-tuned, domain-specific, or reinforcement learning fine-tuned models)?

Q5: Could you investigate the potential for using your metrics to guide the development of better privacy defenses, rather than just evaluating existing attacks?

---

> ### Author Response · Authors · 2025-11-17
>
> We sincerely thank you for the thoughtful comments and constructive suggestions. We have revised the paper accordingly, and all newly added analyses and results have been incorporated into the updated version. Below we summarize your concerns and provide point-by-point responses.
>
> ### **(1) Request for a more detailed explanation of the proposed metrics and their relevance to real-world privacy defense**
>
> **Summary:**
> The reviewer requests deeper clarification on the motivation and usefulness of the proposed metrics, and how they relate to practical privacy protection. (from Weakness 3,5 and  Question 1,3,5)
>
> **Response:**
> Thank you for this thoughtful question. As discussed in Section 1, our metrics are intended as **complements** to dataset-level indicators such as AUC or TPR@lowFPR. Traditional metrics aggregate behavior across an entire dataset, which is useful for summarization but limits interpretability. In contrast, our metrics provided a **sample-specific, fine-grained level** insight into how membership signals arise and how different MIA methods behave on particular instances.
>
> This finer granularity enhances **explainability and diagnostic power**. For example, in Appendix D we show that these metrics reveal why ReCaLL underperforms at low-FPR operating points, which is not visible when relying solely on aggregate statistics. Such analyses help identify method-specific sensitivities and structural failure modes.
>
> Regarding their connection to real-world privacy defense, we view refined MIA evaluation and privacy protection as mutually reinforcing rather than conflicting. The purpose of membership inference research is precisely to understand and mitigate the risks of unintended memorization in LLMs. More nuanced evaluation metrics help uncover the strengths and weaknesses of different MIA techniques, thereby informing better privacy auditing and ultimately contributing to stronger defense strategies.
>
> ### **(2) Concern about how the proposed metrics behave on models beyond Pythia**
>
> **Summary:**
> The reviewer asks whether the proposed metrics generalize to other LLM families and how they would behave when applied outside the Pythia model suite. (from Weakness 2 and Question 2)
>
> **Response:**
> We appreciate the reviewer’s interest in cross-model generality. We would first like to clarify that the proposed metrics are designed to **evaluate MIA methods**, not to evaluate or depend on any particular LLM architecture. This functionality is largely independent of which backbone model is used.
>
> The main reason our experiments focus on the Pythia family is practical rather than methodological. Several MIA methods included in our evaluation make **strong assumptions** like access to the pretraining corpus, which can only be satisfied using models with *fully public* training data. Pythia is one of the few LLM suites that meets these requirements. This reflects the limitations of existing MIA methods themselves, rather than any limitation of our benchmark or our metrics.
>
> From the perspective of the Δ-MIA framework, **transferability to other model families is straightforward**. As clarified earlier, Δ-MIA does not require explicit knowledge of the model’s training set; it requires only that the exposure data be **non-members** of the pre-exposure model. For modern models whose training corpora are unavailable, this can still be ensured reliably by using **post-release timestamped corpora**. For the purposes of this paper, however, evaluating on Pythia is sufficient to benchmark the MIA methods under study, given their specific assumptions and data requirements.
>
> ### **(3) Concern about applying Δ-MIA to various types of post-training models**
>
> **Summary:**
> The reviewer asks how Δ-MIA could be used to assess membership inference risks in models fine-tuned for different purposes, such as instruction-tuned, domain-specific, or reinforcement learning fine-tuned models. (from Question 4)
>
> **Response:**
> We thank the reviewer for raising this thoughtful question. Our work focuses specifically on evaluating **membership inference against the pretraining corpus**, which is the primary privacy concern in large-scale LLMs and aligns with the scope of prior benchmarks such as WikiMIA and MIMIR.
>
> While post-training datasets are important in their own right, attacks targeting these datasets fall under a broader class of problems that extend beyond the scope of the present study. Nonetheless, we emphasize that **the Δ-MIA framework itself has generality**: in principle, the same pre/post comparison could be applied to any training stage, so long as one can obtain a pre-exposure and post-exposure model.
>
> Our decision to focus on the pretraining stage is therefore not a methodological limitation, but a deliberate choice to ensure comparability with existing benchmarks and to maintain a well-defined setting where MIA methods can be fairly evaluated.

---

### Official Review · Reviewer_nrkd · 2025-11-01

**Soundness:** 3
**Presentation:** 3
**Contribution:** 2
**Rating:** 4
**Confidence:** 3

**Summary:**

The paper proposes $\delta$-MIA, an interventional, self-contrast framework that evaluates membership inference by comparing a model’s behavior on the same data before and after controlled exposure, thereby removing cross-dataset distribution shift and enabling sample-level diagnostics (ADR, MVD, EVR, n-MVD). The pipeline logs pre-exposure responses, fine-tunes then stabilizes the model, and computes per-sample deltas; it is instantiated on the Pythia family with The Pile and used to benchmark nine representative MIA methods. Empirically, methods like DC-PDD and Con-ReCaLL decline markedly, while Min-K%++ (and Ref when available) remain strong, with trends strengthening for larger models.

**Strengths:**

1. The paper defines a clear evaluation framework that isolates membership effects by comparing pre- and post-fine-tuning behavior on the same data.


2. The experiments are comprehensive including various MIA probing methods.


3. The introduction of new sample-level metrics (ADR, MVD, EVR, n-MVD) provides additional diagnostic views of membership signals, even if the conceptual novelty is moderate.


4. The paper is generally well organized and readable, with clear visualizations that support its main claims.

**Weaknesses:**

1. The approach requires access to both pre- and post-fine-tuning models (and control the training data to get rid of cross-dataset distribution shift), which may limit practical applicability in real-world privacy audits.
2. The methodological novelty appears limited. The four metrics (ADR, MVD, EVR, n-MVD) appear to be designed as heuristic diagnostics to visualize per-sample score changes before and after exposure.
3. The fine-tuning process isolates the 1.5k target instances, rather than mixing them with non-target samples. Evaluating under more realistic mixed-batch settings would strengthen the validity of the conclusions.

**Questions:**

See weaknesses.

---

> ### Author Response · Authors · 2025-11-17
>
> We sincerely thank you for the thoughtful comments and constructive suggestions. We have revised the paper accordingly, and all newly added analyses and results have been incorporated into the updated version. Below we summarize your concerns and provide point-by-point responses.
>
> ### **(1) Concern about Δ-MIA’s practicality in real-world settings**
>
> **Summary:**
> The reviewer questions whether Δ-MIA can be broadly applied in realistic deployment scenarios, noting that it appears to require access to model weights and may therefore have limited applicability.
>
> **Response:**
> Thank you for raising this important question. We agree that practical applicability is a key requirement for any MIA evaluation framework.
>
> First we would like to emphasize that Δ-MIA does **not** require access to the model’s training data, which is a major limitation of existing train/test-split benchmarks. Instead, Δ-MIA only requires access to **non-member data**, which can be reliably obtained even for closed models by leveraging public model release timestamps to collect post-release corpora. This avoids the unrealistic assumption common in prior benchmarks that the pretraining distribution is available.
>
> Regarding model access, Δ-MIA does not necessarily require access to model internals. What is needed is simply the ability to **perform fine-tuning** and **obtain logits during inference**, both of which are widely supported even by many closed-source LLMs through commercial APIs or managed fine-tuning services. Since Δ-MIA operates entirely at the level of model training interfaces and output distributions, the framework is fully compatible with such environments.
>
> Taken together, the requirements of Δ-MIA on access are strictly **weaker** than those of existing benchmarks. Therefore, Δ-MIA is in fact *more* broadly applicable to real-world privacy auditing scenarios than prior methodologies.
>
> ### **(2) Concern regarding the methodological novelty and the heuristic nature of the proposed metrics**
>
> **Summary:**
> The reviewer questions whether the methodological contribution is substantial and expresses concern that the proposed metrics may be primarily heuristic.
>
> **Response:**
> We appreciate the reviewer’s comments and hope to clarify the contributions of both the framework and the accompanying metrics.
>
> Our methodological contribution lies primarily in addressing two long-standing challenges in MIA evaluation, **distributional shift** and **limited transferability**, by introducing a self-contrastive, intervention-based evaluation framework. Although the mechanism is intentionally simple, it resolves core issues that existing benchmarks have struggled to overcome and enables MIA evaluation in realistic settings where pretraining data are unavailable.
>
> The accompanying metrics are intentionally lightweight but provide complementary value. Existing dataset-level metrics such as AUC or TPR@low FPR provide only coarse global summaries of attack performance. Our metrics offer a **fine-grained, sample-level** decomposition that helps expose where and why different MIA methods succeed or fail. For example, as shown in Appendix D, these metrics help explain ReCaLL’s unexpectedly weak performance at low-FPR operating points, the patterns invisible to aggregate metrics alone.
>
> Thus, although the metrics themselves are simple, they enhance interpretability and diagnostic power in ways not captured by prior evaluations. We hope this clarifies the motivation behind the framework and its supporting metrics.
>
> ### **(3) Question regarding the use of a two-stage fine-tuning procedure**
>
> **Summary:**
> The reviewer asks why target samples are trained in a separate second stage rather than interleaved into a single training run.
>
> **Response:**
> Thank you for raising this question. Our choice of a two-stage fine-tuning design is motivated by the need to maintain **consistent and interpretable recency** across all target data, which is essential for fair comparison of MIA behaviors.
>
> In Δ-MIA, membership evidence arises from differences in the model’s exposure to the target samples. To meaningfully compare per-sample MIA scores, it is important that all target instances have **similar recency** relative to the final model checkpoint. By training on the \~100k target samples as a contiguous block in the second stage, we ensure that their last-seen positions in the optimization trajectory are aligned. This yields a clean and analyzable exposure pattern.
>
> If these target samples were interleaved throughout a longer training sequence, they would be “last seen” at substantially different times. Such variability would introduce recency-induced noise, making some samples appear significantly more or less vulnerable purely due to their placement in the training schedule rather than due to intrinsic MIA effectiveness. This variance would hinder both sample-level interpretability and the fairness of cross-method comparison.

---

### Official Review · Reviewer_wA5G · 2025-11-03

**Soundness:** 2
**Presentation:** 3
**Contribution:** 3
**Rating:** 6
**Confidence:** 4

**Summary:**

This paper proposes a new evaluation framework, Delta-MIA, for evaluating MIA performance on LLMs to tackle three major limitations of typical observational evaluation framework: 1) unintended distribution shift due to challenges with partitioning member/non-member data (e.g., temporal shifts) resulting in inflated performance, 2) lack of coarse-grained analysis, obscuring sample-level behavior, and 3) poor transferability due to reliance on training corpora access. The authors instead propose an alternative perspective using an interventional paradigm. The main idea is to measure pure membership signals by performing a self-comparison between a model before and after exposure to some sample data using scores from a candidate MIA. The authors also introduce new sample-level metrics (e.g., ADR, MVD, EVR, and n-MVD) that allow for more granular analysis over captured membership signals. They then conduct a broad evaluation over nine modern MIAs on subsets of Pile data over the Pythia model family. Most MIA methods remain robust, but certain methods such as DC-PDD and Con-ReCaLL are shown to not capture membership signals as well as previously claimed.

**Strengths:**

- Delta-MIA tackles several important challenges in MI evaluation that are crucial for validating the effectiveness of current and future MIAs. The paper is well-written, easy to follow, and the problem is clearly motivated.
- The proposed framework is conceptually-straightforward and the introduced metrics are also intuitive. The authors also take care to clearly interpret visualizations to help readers understand nuanced performance differences between the different MIAs.
- The evaluation is conducted over a broad range of models, MIAs, and data domains.

**Weaknesses:**

- This framework is a useful, necessary check to ensure that a candidate MIA is not capturing spurious signals (e.g. from unintended distribution shifts between benchmark members/non-members) and can actually detect true membership signals. However, I’m unconvinced this framework is sufficient to determine the effectiveness (i.e., vulnerability risk) of an MIA. Unless I have a misunderstanding, without comparing MIA signals between members and non-members under the same model, the discriminatory power of the MIA isn’t interpretable. For example, what if the MVD (or n-MVD) is also high for a set of non-members (to both the pre-/post-exposure models) from the same distribution as the target data? Then the model may not truly be that vulnerable to the MIA and the signal being captured may be something more than just membership. Perhaps it would be stronger to more clearly frame it as a complementary method (e.g., sanity check) to standard observational evaluation? If this is not the case, then it is still a little unclear to me how delta-MIA would be sufficient on its own.
- This framework also seems to be heavily dependent on the tuning process. I feel that there could be more discussion about the impact of the tuning phase, such as the choice of target tuning data (e.g., what domains), how much target data is used, and other design choices. Ablations in these and similar directions would be appreciated.
- Closely related to the above comment, it’s not clear to me how this framework bypasses the issue of choosing non-member samples. For example, target tuning data still needs to be verified as non-members to the pre-exposure model, which remains difficult for modern frontier models.

I include more specific questions related to these points in the question section.

**Questions:**

- Could the scores presented still be inflated due to the recency of the target data relative to the entire training lifecycle? What if the injected data was instead inserted, for example, halfway through training. Currently, the scope seems more like “evaluating MIAs on finetuned LLMs”.
- Similar to the subexperiment in Appendix A, could the authors show how the tuning impacts performance on data from roughly the same distribution (e.g., another sub-sample from Pile test for Pile-CC, Wikipedia, etc.)?
- Using the post-exposure model as the target model, could the authors also conduct a standard observational evaluation (using the target tuning data as members and another random sample of Pile test data selected the same way as non-members). It’d be interesting to concretely see if performance trends in this observational setting align with those under delta-MIA (e.g., maybe some attacks still seem performant in this observational setting, but under delta-MIA aren’t).
- What is the reason for having two-stage finetuning? For example, why not batch/randomize the selected 1500 samples with the 100000 samples, training in one stage?
- Do the authors have any results on different model families (e.g., Llama) to demonstrate the transferability of their approach?

Minor comments:
In Figure 1, target is misspelled as “traget”

---

> ### Author Response · Authors · 2025-11-17
> **Part 1/2 of the Response to  Reviewer wA5G**
>
> We sincerely thank you for the thoughtful comments and constructive suggestions, especially for the careful reading of our paper. We have revised the paper accordingly, corrected the typos, and all newly added analyses and results have been incorporated into the updated version. Below we summarize your concerns and provide point-by-point responses. Due to space limitations, we will respond in two parts.
>
> ***
>
> ### **(1) Concern about robustness of Δ-MIA when evaluating distributionally similar non-members**
>
> **Summary:**
> The reviewer questions whether Δ-MIA remains reliable when the non-member data come from the same distribution as the target exposure data and suggests including experiments comparing model behavior before and after fine-tuning under matched-distribution settings. (from Weakness 1 and Question 2)
>
> **Response:**
> Thank you very much for this insightful suggestion. We conducted an additional empirical analysis to examine this scenario. Below is a brief summary table of the experimental results. Detailed results and analysis have been added to Appendix A.
>
> |Category|KL Mean|KL Median|Loss (Before Fine-Tuning)|Loss (After Fine-Tuning)|
> |--|--|--|--|--|
> |PubMed Abstracts|0.1707|0.1608|2.1280|2.1920|
> |Pile-CC|0.1706|0.1544|2.6497|2.7593|
> |Wikipedia (en)|0.2048|0.1844|1.8947|2.0023|
> |**Overall**|**0.1820**|**0.1650**|**2.2241**|**2.3179**|
>
> In the new experiment, we randomly sampled 1,500 samples from the same three domains of **Pile validation split** (500 per domain). For each sample, we computed the per-sample averaged KL divergence between the vocabulary probability distributions of the two models. The results show that these KL divergences are tightly **concentrated around zero**, indicating that the two models produce highly similar output distributions on such non-member data.
>
> This observation directly supports the **robustness** of Δ-MIA. Since mainstream MIA methods infer membership by detecting changes in output distributions across models, non-member samples whose distributions remain nearly unchanged across pre- and post-exposure models naturally lead to nearly identical MIA scores, and therefore cannot spuriously produce high MVD values.
>
> ### **(2) Request to evaluate MIA methods on the post-exposure model under the standard observational paradigm**
>
> **Summary:**
> The reviewer suggests conducting an additional observational evaluation using the post-exposure model as the target in order to compare performance trends between Δ-MIA and the traditional observational MIA setting. (from Question 3)
>
> **Response:**
> We appreciate this insightful suggestion. Following the reviewer’s recommendation, we have added a supplementary observational evaluation using the post-exposure Pythia models. This new experiment and its full results are included in Appendix E of the revised manuscript. Here we merely show the performance of MIAs on Pythia-post-2.8B.
>
> |Method|AUC|T@5%F|
> |--|--|--|
> |Loss|0.69|0.11|
> |Zlib|0.66|0.20|
> |Neighborhood|0.64|0.10|
> |Con-ReCaLL|0.46|0.061|
> |DC-PDD|0.75|0.24|
> |Min-K%|0.74|0.15|
> |ReCaLL|0.70|0.10|
> |Ref|0.82|0.35|
> |Min-K%++|0.85|0.35|
>
> Using target dataset as members and another 1,500 validation samples from the Pile dataset as non-members, we evaluate all MIA methods on the Pythia-post models following the standard observational protocol. For most methods, including the strongest performers under Δ-MIA, such as Min-K%++ and Ref, as well as baselines such as Loss, the observational results closely mirror their Δ-MIA performance, demonstrating consistent relative rankings across paradigms.
>
> Two methods show noticeable deviations, consistent with their design characteristics. ReCaLL performs substantially worse in the observational setting, likely due to its sensitivity to prefix-data distributions: in Δ-MIA, members and non-members share identical inputs (with only the model state changing), eliminating distributional confounders; in observational evaluation, small differences of data can reduce the effectiveness of prefix data, especially without hyperparameter tuning (which we deliberately avoid to ensure fair comparison). Conversely, DC-PDD achieves stronger results in the observational setting but performs poorly under Δ-MIA, which aligns with our expectation: DC-PDD relies on token frequency statistics from external corpora and is therefore more susceptible to exploiting subtle distributional differences that Δ-MIA explicitly eliminates by construction.
>
> Overall, the strong consistency observed across most methods supports the alignment between Δ-MIA and the traditional observational paradigm, while the discrepancies for ReCaLL and DC-PDD are readily explained by their inherent sensitivities rather than by differences in evaluation frameworks.

---

> > ### Author Response · Authors · 2025-11-17
> > **Part 2/2 of the Response to Reviewer wA5G**
> >
> > ### **(3) Clarification of fine-tuning design choices**
> >
> > **Summary:**
> > The reviewer raises questions about (1) how target domains and data quantities were selected, (2) why we adopt two-stage rather than interleaved fine-tuning, and (3) whether the smaller training scale may inflate MIA scores. (from Weakness 2 and Question 1,4)
> >
> > **Response:**
> > While Section 2.2 and Appendix B.2 provide partial explanations, we agree that additional clarification is helpful, and we have provided further explanations below.
> >
> > **Target data and hyperparameter choices.**
> > Our target-domain selection generally follows MIMIR with two adjustments. First, we exclude the GitHub subset from Pile because it contains substantial non-English and malformed content (e.g., URL fragments), which destabilize MIA evaluation. Second, because several post-MIMIR MIA methods (e.g., ReCaLL) rely on non-member prefixes and are sensitive to available context, we choose datasets whose sample lengths remain within passage-level to ensure consistent applicability across methods.
> >
> > For other fine-tuning hyperparameters, our principle is to remain as close as possible to the pretraining regime. The choice of \~100k samples (≈10⁸ tokens) is inspired by MIMIR’s analysis of *training step* (recency) effects: this scale is sufficient to move the target samples past the short-term spike that occurs immediately after first exposure, while keeping their overall proportion (≈1.5%) small enough to maintain stable behavior.
> >
> > **Rationale for two-stage fine-tuning.**
> > As clarified in the revision, the two-stage setup provides precise control over *recency*, which is critical for interpretable MIA comparison. Training on ∼100k samples in a contiguous block ensures that all target instances share a similar distance from the final optimization step. In contrast, interleaving them throughout a longer training trajectory would cause different target samples to be last seen at highly uneven times, introducing recency-induced variance and reducing the comparability of per-sample MIA scores.
> >
> > **On whether MIA scores may be inflated.**
> > We agree that absolute MIA scores can be higher than those observed in full-scale pretraining, because our shorter training trajectory positions target samples at relatively higher recency. However, absolute parity with full pretraining is not the goal of Δ-MIA. Our objective is to establish a controlled setting where *relative* comparisons across models and MIA methods remain fair. The ∼100k-sample second stage mitigates the extreme “just-seen” effect while preserving a consistent recency configuration for all evaluated methods, ensuring that any systematic upward shift affects all methods equally and does not alter their relative ordering.
> >
> > Moreover, as shown by prior work on pretraining forgetting \[1], some samples in large-scale training corpora are naturally forgotten. MIA is fundamentally motivated by privacy risk assessment and is only meaningful on data that the model truly retains. Forgotten samples, even if labeled “members”, do not pose privacy risk, and thus poor MIA performance on them is expected and not necessarily indicative of attack failure. This phenomenon may partially explain why benchmarks such as MIMIR sometimes report uniformly low MIA scores, which can underestimate the underlying capability of modern MIA methods.
> >
> > \[1] Exploring Forgetting in Large Language Model Pre-Training (ACL 2025)
> >
> > ### **(4) Clarification on the transferability of Δ-MIA**
> >
> > **Summary:**
> > The reviewer expresses uncertainty about how Δ-MIA applies to other model families (e.g., Llama) and raises the concern that it is unclear how the framework avoids the challenge of selecting non-member samples. (from Weakness 3 and Question 5)
> >
> > **Response:**
> > We mentioned this in Section 1, but our wording may not have been accurate. We would like to clarify that there are some misunderstandings here.
> >
> > Δ-MIA does **not** aim to avoid selecting non-member samples; instead, it avoids requiring *explicit knowledge of member samples*. The key requirement is simply that the target data used for exposure must be **non-members with respect to the pre-exposure model**. For modern frontier models whose training sets are typically unavailable, this can still be achieved reliably by leveraging known model release timestamps. Following the strategy used in WikiMIA, one can collect data from sources such as Wikipedia or news corpora published *after* the model’s release time, ensuring that the pre-exposure model could not have seen these samples during pretraining.
> >
> > In our work, we use Pythia primarily because several existing MIA methods impose additional assumptions, such as access to the training corpus, which can only be satisfied on fully open-source models with publicly available training sets. This is a limitation of the MIA methods themselves, not of Δ-MIA. From the perspective of the benchmark design, Δ-MIA is readily transferable to other model families.

---

### Author Response · Authors · 2025-11-24
**Summary of Rebuttal**

We regret that, due to the unexpected changes in conference procedures, we were unable to receive additional reviewer comments during the discussion period. Nevertheless, we appreciate that several reviewers provided thoughtful insights and recognized the significance of our work. We also identified that certain concerns stemmed from misunderstandings, all of which we have addressed comprehensively in our rebuttal. Below is a concise summary of our responses:

1. **Response to Reviewer wA5G**

We are pleased that Reviewer wA5G acknowledged the value of our work and its critical role in advancing the field, stating,
>**“Delta-MIA tackles several important challenges in MI evaluation that are crucial for validating the effectiveness of current and future MIAs.”**

This reviewer carefully engaged with our paper, even identifying typos, and raised high-quality concerns. To address these, we added two sets of experiments: (1) demonstrating the **robustness** of the Δ-MIA framework when non-member data are distributionally similar to target data; and (2) conducting an observational evaluation using the post-exposure model, empirically validating the **equivalence** between Δ-MIA and traditional MIA evaluation settings. These experiments enhance the **completeness** and **reliability** of Δ-MIA.

We also addressed other concerns, including detailed explanations of the fine-tuning design and its validity (section (3)) and clarifications on the **transferability** of Δ-MIA (section (4)).

2. **Response to Reviewer nrkd**

Reviewer nrkd’s primary concern centered on the practical applicability of our work in real-world privacy auditing. In reality, Δ-MIA has **strictly stronger practicality** than prior benchmarks: it **eliminates the critical limitation** of requiring access to model training data (largely unavailable in practice) and **only necessitates** fine-tuning capabilities and logit access, satisfied by both open-source models and many closed-source models via commercial APIs (detailed in Section (1)).

We also addressed other concerns: (2) emphasized the key role of our proposed metrics in MIA **diagnosis and analysis**, supported by Appendix D; and (3) clarified the **well-motivated necessity** of two-stage fine-tuning, resolving potential misunderstandings.

3. **Response to Reviewer 1iHS**

Some of Reviewer 1iHS’s questions might be broad and somewhat peripheral to the core challenges typically examined in membership inference research. Nevertheless, we responded thoroughly and constructively to each point.

In section (1) we explained the relevance of our metrics to real-world privacy assessment and how they complement dataset-level measures by offering **fine-grained, interpretable diagnostics**. We also (2) clarified that the metrics evaluate MIA methods and are **not tied to any specific LLM family**, with Pythia chosen solely to match the constraints of existing MIA implementations, and (3) elaborated on the generality of Δ-MIA beyond pretraining-stage evaluation, while noting that we focused on pretraining to maintain **consistent comparison with prior benchmarks**, ensuring fairness and clarity.

4. **Response to Reviewer nYoz**

Reviewer nYoz demonstrated strong familiarity with related work but still held several misunderstandings regarding Δ-MIA’s motivation and behavior. Importantly, the reviewer noted that they would consider **raising their score** if their concerns were resolved; although procedural changes prevented further dialogue, we provided comprehensive clarifications.

In section (1), we explained that Δ-MIA is designed to **ensure fairness** in MIA evaluation, eliminate distribution shift, and provide finer-grained analytical insights. We clarified why Δ-MIA may yield higher absolute MIA scores while still supporting fair comparative evaluation. Meanwhile, recent findings on LLM pretraining forgetting (section (4)) indicate that prior observational benchmarks may **systematically underestimate** MIA performance. We also clarified Δ-MIA’s **compatibility** with fine-tuning-based MIAs in section (2), **corrected misunderstandings** regarding data selection in section (3), and thoroughly addressed concerns about Δ-MIA’s **reliability** (section (4)). The two additional experiments added in response to Reviewer wA5G directly support these clarifications by empirically validating Δ-MIA’s **robustness** and its **alignment with observational MIA settings**.

---

### Meta-Review · Area_Chair_SoWg · 2026-01-04

**Summary:**

The submission proposes a new evaluation framework for membership inference attacks (MIAs). This is a dataset and benchmark track paper.

The key idea is to identify true membership signals by comparing how a model responds to data before and after being trained on it. This self-comparison approach uses the same sample in two ways: first as a non-member (before the model has seen it) and then as a member (after training / fine-tuning on it). By measuring the difference in the model's response between these two states, the proposed evaluation MIA framework can isolate the actual effect of membership.

The paper makes extremely strong assumptions that are met in practice for only very few frontier models (namely two that we know of: Pythia and OLMo). Specifically, the paper states as the first stage of the proposed framework: "(1) Collect a set of unseen data with verifiable non-overlap with the training corpus of the target model, and
record responses in the pre-exposure state". This requirement can be met only with open-source models such as Pythia and OLMo. However, these are the only models on which the framework can be applied. This is big drawback that the evaluation in the paper itself is limited solely to the Pythia suite of models.

This paper does not consider the state-of-the-art evaluation of MIAs against LLMs, for example, [1]. The crucial point is that we know how to evaluate MIAs for the fine-tuning of LLMs (including strong MIAs such as LiRA or RMIA), however, the evaluation of MIAs for pre-training is the current main problem, since we often lack the access to the ground truth training (and validation) data and the shadow models are non-existent.

The authors introduce three delta metrics but there is no motivation behind why we need these metrics and where they stem from. "These metrics enable visual and quantitative tests of key MIA hypotheses." This statement in the intro does not help much. There is only a single MIA null hypothesis: "A target point is a non-member" (e.g., Carlini et al., 2021).

For the paper in the dataset and benchmark track, the code and data should be available as an anonymous repository.

Regarding the "Practicality and applicability to real-world LLMs" - this framework is not applicable to the commercial APIs since the training sets of the LLMs behind the APIs are unknown. The proposed framework requires the Target Data (Unseen) that was not used to train the target model. Furthermore, as indicated by reviewer 1iHS "The paper lacks sufficient comparison with existing MIA evaluation frameworks, making it difficult to fully appreciate the novelty of the proposed metrics." Finally, Reviewer nYoz points out that: "the paper does not theoretically or empirically establish equivalence between the Delta-MIA setting (before/after fine-tuning on a held-out dataset) and the conventional MIA problem (evaluating a pretrained model on candidate data with no distribution shift)".

Overall, there is a consensus among the reviewers that the paper should be rejected. I agree with this assessment.

**References:**

[1] Hayes et al. "Exploring the limits of strong membership inference attacks on large language models" NeurIPS 2025 https://arxiv.org/abs/2505.18773 (available since May 24th 2025).

**Reviewer Concerns:**

Overall, the concerns raised by reviewers were not fully addressed in the rebuttal period.

One of the main issues are:

1. Reviewer 1iHS: "The paper lacks sufficient comparison with existing MIA evaluation frameworks, making it difficult to fully appreciate the novelty of the proposed metrics."

2. Reviewer nYoz points out that: "the paper does not theoretically or empirically establish equivalence between the Delta-MIA setting (before/after fine-tuning on a held-out dataset) and the conventional MIA problem (evaluating a pretrained model on candidate data with no distribution shift)".

**Reviewer Scores:**

Reviewer nrkd: Score 4 / Confidence 3

Reviewer i8z9: No Review

Reviewer 1iHS: Score 4 / Confidence 4

Reviewer nYoz: Score 4 / Confidence 5

Reviewer wA5G: Score 6 / Confidence 4

---

> ### Public Comment · ~Xuewei_Yang1 · 2026-02-27
> **clarification from authors**
>
> As one of the authors of this paper, I regret to see that the Area Chair appears to have significant misunderstandings about our work. To prevent potential readers who may visit this page from being misled, I would like to offer the following brief clarifications:
>
> Most serious misunderstanding: The Area Chair mistakenly interpreted a key contribution of our work—addressing a known limitation in prior work—as a weakness of our own approach. Specifically, they claimed that our Delta-MIA framework relies on an unrealistically strong assumption requiring access to the LLM’s training data. This is incorrect. As demonstrated in [1], unseen data can be verified via timestamps after the model's release date without requiring any access to member data. Moreover, our choice to evaluate using the Pythia model suite was made solely to ensure compatibility with specific MIA evaluation protocols.
>
> Explanation of the three delta metrics: These metrics naturally arise from our self-contrast design. While simple and intuitive, they are highly informative. In both the Introduction and the Metric Design section, we clearly explain what each metric captures—namely, detectability, separation, and noise. Furthermore, Appendix D provides concrete examples showing how these metrics can be used to interpret and analyze specific MIA methods.
>
> During the rebuttal period, we thoroughly responded to all reviewer concerns and provided additional clarifications and results in the appendix. Unfortunately, it seems the Area Chair did not take these responses into account.
>
> [1] Shi et al. "Detecting Pretraining Data from Large Language Models" ICLR 2024 https://arxiv.org/abs/2310.16789

---

### Decision · Program_Chairs · 2026-01-26

Reject